# The dependence of children's generalization on episodic memory varies with age and level of abstraction

Sabrina Karjack [1] ✉, Nora S. Newcombe [2] & Chi T. Ngo [3] ✉

Generalization extends learning from specific to new examples, whereas episodic memory preserves specific instances. Some models suggest generalization occurs by retrieving individual but related episodes, a pattern found in young adults. Children's generalization of characteristics of individual people may rely instead on semantics. Here, we study generalizations across multiple levels of abstraction and characterize their contingency on episodic memory. Children (3-8 years, N = 121) watch animals find homes in different places. These events contain patterns of regularities linking different hierarchical levels of places and animal species. We assess children's inferences about unstudied animal-place associations at various abstraction levels and memory precision for individual episodes. We find that lower-level generalization depends on memory of specific episodes, and the degree of dependence increases with age. In contrast, higher-level generalization is not statistically associated with episodic memory. This work shows that the contingency of generalization on memory specificity depends on age and level of abstraction.

Imagine that on a scuba diving trip, you meet several sea otters along your route in a spectacular kelp forest. Later on that trip, you notice a few eye-catching giant clams while exploring a coral reef. Memories for experiences like these, often referred to as *episodic memories*, make up a sense of our personal history and (sometimes) serve as a veridical record of our past[1]. Remembering the place in the kelp forest where you spotted the brown sea otter is an example of such an episodic memory. In addition, memory for these events also contributes to our semantic knowledge about the world—knowledge that is often built from the statistical regularities across many experiences[1]. Generalization applies such knowledge in new situations. For example, you may expect an encounter with sea otters in a kelp forest or see clams occupying a coral reef on your next snorkeling excursion. In fact, you may even suspect that other sea mammals, such as seals, can also be found around kelp forests, and that other mollusks, such as oysters, would find their homes on coral reefs. Memory jointly subserves our abilities to preserve the specific instances of the past and to adaptively generalize in new circumstances.

How do our episodic memories contribute to the initial acquisition of semantic knowledge that enables generalization in new instances with varying degrees of abstraction from the directly learned experiences? Some models propose that new information is originally stored as hippocampus-dependent episodic memory, which later transforms into neocortical semantic memory through repetition or gist extraction[2,3]. They also suggest a neocortical pathway for slowly acquiring semantic memory that may bypass the hippocampus. However, generalization can happen rapidly, after hours or even minutes of exposure to a recurring pattern. On this timescale, rapid generalization may, in part, rely on the retrieval of individual yet related episodes[4,5]. If so, generalization should be contingent on successful episodic memory retrieval, a pattern that has been observed in young adults[6–8].

The interface between episodic memory and generalization is particularly interesting in the context of early development. Semantic memory develops early in life: infants accrue knowledge about their environment and can generalize past experiences over new stimuli and context[9–12]. Infants build an impressive vocabulary[13], and begin to

[1]Department of Psychology, University of California, Davis, Davis, CA, USA. [2]Department of Psychology, Temple University, Philadelphia, PA, USA. [3]Center for Lifespan Psychology, Max Planck Institute for Human Development, Berlin, Germany. ✉e-mail: skarjack@ucdavis.edu; ngo@mpib-berlin.mpg.de

group things into categories[14]. Between the first and the second year of life, they can imitate short sequences performed on one set of stimuli with a new set of stimuli[15,16]. In contrast, episodic memory is not evident until the end of the second year, and follows a slower trajectory throughout childhood[17]. Both aspects of episodic memory – remembering what happened, associated with where it had happened, *and* memory precision for the details of such events[18] – improve from early to middle childhood[17]. Many studies show that children's associative memories between places and objects or actions improved from ages 4 to 8 or 9[19,20]. We know less about memory precision in early development because only a handful of studies have quantified memory precision as the deviation between true and retrieved memories in young children. These studies show that memory precision for the specific objects' positions within a spatial array improves from ages 3 to 7[21] or even into adulthood[22,23]. In sum, younger children are less able to bind items to locations and reconstruct the item-position associations with less precision.

The uneven development of semantic memory and episodic memory raises the question of how the relation between rapid generalization and episodic memory unfolds over the early years of life. Which aspects of specific episodes support rapid generalization in young children, and does the answer depend on age? Findings from two studies suggest that the semantic structure of past experiences may promote generalization from early to middle childhood rather than episodic memory[7,24]. Using the same paradigm, these two studies tested generalization and memory specificity, with memory specificity being assessed for multiple aspects of the learned episodes. Participants were shown many characters, each of whom went to four different but semantically-related places (e.g., an art museum, an art studio) and collected a semantically-congruent object (e.g., a canvas, paint brushes) at each place. Participants were then asked to generalize about each character by selecting an unstudied object that belonged to the same semantic category as those paired with that character during encoding. Subsequently, memory specificity for different aspects of learned episodes was assessed, including the context (which object was collected where), object identity (which of the art-related objects was collected), and object perceptual details (which of the canvas exemplars was seen at encoding).

The key finding from this small line of work is that generalization accuracy in adults, but not in children, is contingent on their memories for the context bound to a given object[7]. It is likely that with development, the strengthening of episodic memory capacity impacts the degree to which detailed memories of past instances support generalization. Children, instead, rely on their memories for the conceptual aspects of the objects. That is, memory specificity for the object identity (e.g., remembering seeing a canvas, not a sketch book) is associated with better generalization performance. In addition, the semantic proximity among objects within the same category matters for children's generalization, with greater proximity resulting in better generalization accuracy[7,24]. These findings suggest that the semantic structure underlying a set of related experiences is important for generalization in early childhood.

In these previous studies, the semantic categories are individual object categories, such as art-related objects versus musical instruments related to individual cartoon characters, e.g., Moomin. However, semantic knowledge encompasses interconnected networks and hierarchies of concepts[25]. Young learners begin to form basic-level categories, such as dogs versus cats, in infancy, with evidence in language production by around 18–24 months[26–28]. However, children's understanding that dogs and cats are part of a larger group called mammals, distinct from other groups of animals like birds, may not come until 3 or 4 years old[29]. The development of superordinate categorization builds on the foundation of basic-level categories, and is marked by a more abstract understanding of how basic-level categories form broader sets of concepts[30]. To date, very little is known about children's abilities to generalize to new instances at varying levels of the categorical memberships and how such abilities differ across childhood. Further, we do not know how different levels of generalization, reflecting varying levels of abstraction, are tethered to episodic memories, and whether these linkages vary across childhood.

Generalization enables us to extract rules from specific experiences and apply them not only to new examples (low-level generalization) but also to broader conceptual principles (high-level generalization). Departing from previous studies that focused on the contingency of low-level generalizations on episodic memory, we modeled generalizations that involve the deployment of knowledge across multiple levels of abstraction. Importantly, we examined how this contingency varies between ages 3 and 8, a window marked by substantial growth in episodic memory capacities[17].

Our first aim was to characterize age patterns in children's ability to generalize across varying levels of categorical membership, based on regularities drawn from specific instances. We hypothesized that with age, children would show improved generalization performances, consistent with previous findings on the development of generalization. Second, we asked whether successful generalization depends on the memory specificity of the individual instances, and if so, whether this relationship varies by age and level of abstraction. Low-level generalization involves applying knowledge to new instances that closely resemble directly learned experiences, whereas high-level generalization involves abstraction over broad category structures. We expected a significant age-related increase in the coupling between generalization and memory specificity at the lower level, aligning with previous studies[7,24]. However, it remained unclear whether intermediate- and high-level generalization would similarly depend on episodic memory. One possibility is that all forms of rapid generalization rely on episodic memory. Alternatively, high-level generalization may be independent of the precision with which children retrieve episodes.

Taking inspiration from the taxonomy of superordinate and basic-level categorization[25,31,32], we assessed multi-level generalization with the Home Sweet Home memory game. In this paradigm, we also examined children's memory specificity of the individual episodes. Children watched a series of animals (e.g., Peggy the horse), each finding a home in one of two new towns. We embedded two nested levels of regularities. First, animal species belonging to the same superordinate-level category were assigned to the same town, such that mammals were placed in one town (e.g., Rubyville), and birds in the other. Second, members of the same basic-level category (e.g., horses) were assigned to one of the four locations within a town (e.g., the castle in Rubyville). Each individual member of a species occupied a unique position within that location (e.g., Peggy was to the left of the castle). We then tested whether children could successfully generalize at multiple levels, ranging from specific animal species (e.g., a new horse, a baby horse) to broader classes of animals (e.g., an unstudied mammal species: a pig) (see Fig. 1 for a conceptual depiction of the task design). We then linked generalization performances to children's memory precision for the location assigned to each individual animal. This study was not pre-registered.

Here, we show that generalization across multiple categorical levels improves drastically from early to middle childhood. The contingency of generalization on memory specificity depends on both levels of abstraction and the age of the generalizer. Higher-level generalization is unanchored to memory specificity, whereas lower-level generalization relies on it. The association between low-level generalization strengthens with age. Finally, younger children's weaker contingency of generalization on memory specificity stems from their ability to generalize accurately despite having unreliable memory for specific instances, rather than the reverse.

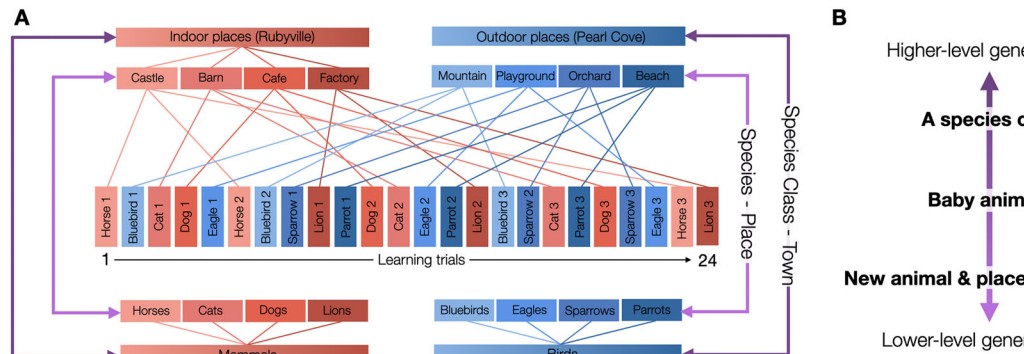

**Fig. 1 | A conceptual depiction of the task design. A** In each block of 24 learning trials, children saw three exemplars (e.g., horse 1, horse 2, horse 3) of each animal species (e.g., horses) moving to a specific place (e.g., castle) in a specific town (e.g., Rubyville) in an interleaved fashion. All animal species that moved to Rubyville were mammals (e.g., horses, cats, dogs, lions), whereas all species that moved to Topaz Valley were birds (e.g., bluebirds, eagles, sparrows, parrots). **B** A set of generalization tasks was designed to estimate children's abilities to generalize based on the specific learned animal-place associations, spanning low-, intermediate-, to high-

level generalizations. For lower-level generalization, children were asked to apply the learned associations to new exemplars of both animals and places. For intermediate-level generalization, they extended this knowledge to baby animals that, by design, appeared less similar to those seen during learning. For high-level generalization, children applied their knowledge to entirely new species that belonged to the same broader class, mammals or birds, as those encountered during learning.

## Results

There were no statistically significant sex differences on any of the generalization tasks (New Animals: $t(117) = 0.08$, $p = 0.938$, 95% CI [−0.105, 0.114], New Places: $t(117) = 0.07$, $p = 0.941$, 95% CI [−0.107, 0.115], Baby Animals: $t(117) = 0.11$, $p = 0.914$, 95% CI [0.756, 0.750], New Species: $t(117) = 0.50$, $p = 0.619$, 95% CI [0.629, 0.607]), or on memory specificity (Displacement error: $t(114) = −0.793$, $p = .430$, 95% CI [0.257, 0.288]; Regional accuracy $t(117) = 0.713$, $p = 0.477$, 95% CI [0.837, 0.805]). Inter-task correlations are reported in Supplementary Note 2 (see Fig. S1). To examine age-related differences in generalization while accounting for individual differences in verbal ability, PPVT was included in all models. We also tested whether generalization performance differed between typical and atypical animal species by including typicality as a fixed effect. It is possible that generalization performance, particularly in the New Species task, would be more robust for typical compared to atypical animal species[33,34]. For consistency, typicality was included in all models that predicted generalization performance.

### Age-related improvements in generalization accuracy

First, we tested whether generalization accuracy was associated with children's age (measured in months), verbal intelligence (standardized PPVT score), and typicality. We conducted a series of generalized mixed-effects models, one for each generalization task. For New Animals, New Places, and Baby Animals generalizations, the model was specified as follows: lmer(generalization accuracy ~ age * typicality + verbal intelligence + (1 | participant)).

Similar patterns emerged across the New Animals, New Places, and Baby Animals tasks: both age (see Fig. 2) and verbal intelligence, but not typicality, were significantly associated with generalization accuracy. For New Animals generalization, accuracy was positively associated with age, $\beta = 0.0085$, $SE = 0.0012$, $t(183.4) = 7.11$, $p < 0.001$, 95% CI [0.006, 0.011], and verbal intelligence, $\beta = 0.004$, $SE = 0.0019$, $t(118.2) = 2.03$, $p = 0.044$, 95% CI [0.0001, 0.007]. Typicality, either as a main effect, $\beta = −0.008$, $SE = 0.08$, $t(119.0) = −0.10$, $p = 0.922$, 95% CI [−0.173, 0.157], or in interaction with age, $\beta = 0.0005$, $SE = 0.001$, $t(119.0) = 0.41$, $p = 0.683$, 95% CI [−0.002, 0.003] showed no statistically significant effect on generalization accuracy. For New Places generalization, accuracy was again positively associated with age, $\beta = 0.0085$, $SE = 0.0012$, $t(195.1) = 7.27$, $p < 0.001$, 95% CI [0.006, 0.011], verbal intelligence, $\beta = 0.004$, $SE = 0.0018$, $t(118.0) = 2.50$, $p = 0.014$, 95% CI [0.001, 0.008]. Typicality, $\beta = −0.0645$, $SE = 0.09$,

$t(119.0) = −0.72$, $p = 0.472$, 95% CI [−0.239, 0.111], or its interaction with age, $\beta = 0.001$, $SE = 0.001$, $t(119.0) = 1.13$, $p = 0.263$, 95% CI [−0.001, 0.004], showed no statistically significant effect on performance. In the Baby Animals task, generalization accuracy was significantly associated with age, $\beta = 0.01$, $SE = 0.001$, $t(180.6) = 8.90$, $p < 0.001$, 95% CI [0.008, 0.012], and showed a trend toward a significant association with verbal intelligence, $\beta = 0.003$, $SE = 0.002$, $t(180.0) = 1.90$, $p = 0.060$, 95% CI [−0.0001, 0.007]. Again, typicality showed no statistically significant effect, either as a main effect, $\beta = 0.05$, $SE = 0.08$, $t(190.0) = 0.71$, $p = 0.479$, 95% CI [−0.095, 0.204], or in interaction with age, $\beta = 0.00008$, $SE = 0.001$, $t(190.0) = 0.08$, $p = 0.939$, 95% CI [−0.002, 0.002]. These results suggest that age, above and beyond interindividual differences in verbal intelligence, positively scaled with children's ability to generalize to new places, new animal exemplars, and baby animals within a species.

For the New Species task, we included species class knowledge as an additional fixed effect, as this task required generalizing across species within the same class (i.e., mammals or birds, see also Fig. S3). The model was specified as: glmer(generalization accuracy ~ age * typicality + verbal intelligence + species class category knowledge + (1 | participant)). There was no statistically significant effect of verbal intelligence, $\beta = 0.002$, $SE = 0.002$, $t(117.0) = 1.35$, $p = 0.178$, 95% CI [−0.001, 0.006], nor species class category knowledge, $\beta = −0.02$, $SE = 0.12$, $t(117.0) = −0.17$, $p = 0.865$, 95% CI [−0.254, 0.213]. Generalization accuracy was again positively associated with age, $\beta = 0.008$, $SE = 0.001$, $t(191.97) = 5.95$, $p < 0.001$, 95% CI [0.005, 0.010], which interacted with typicality, $\beta = −0.004$, $SE = 0.001$, $t(191.0) = −2.82$, $p = 0.006$, 95% CI [−0.006, −0.001]. This interaction showed that the age-related improvements in New Species generalization were significantly greater for typical species compared to atypical ones (see Fig. 3).

Together, these results suggest that with age, children were better at forming new generalizations for each element of the learned associations, including both animals and places. They were also better at bringing their knowledge to bear on new inferences to less perceptually similar members within a given species and even more broadly, to previously unstudied species within a species class. Notably, older children were particularly better than younger children at generalizing to new species that were typical members of their species class. This age-related improvement was less pronounced for atypical species.

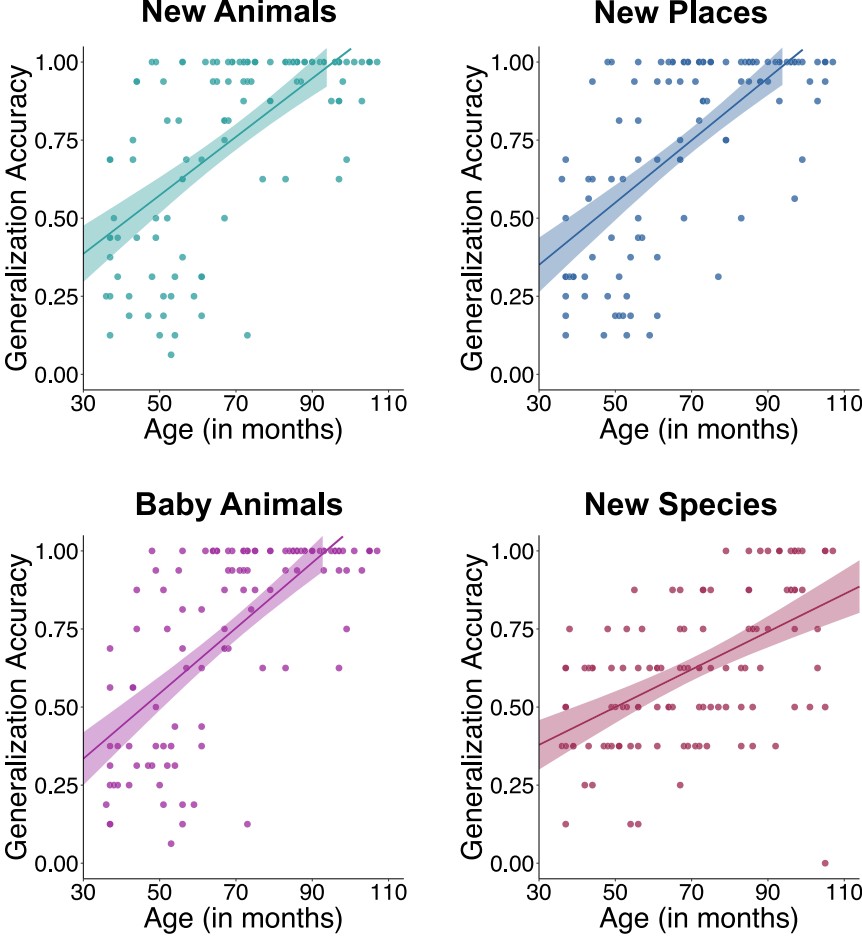

**Fig. 2 | Age patterns in generalization performances separated by tasks.** Scatterplots of bivariate Pearson correlations between age (measured in months, on the x-axes) and generalization accuracy shown (on the y-axes) for the New Animals (shown in turquoise), New Places (shown in navy), Baby Animals (shown in purple), and New Species (shown in maroon) tasks. Shaded ribbons represent ±1 standard error around the linear regression line in each plot.

## Age-related patterns in generalization error

Next, we examined children's generalization failure, or error. In the New Animals, New Places, and Baby Animals tasks, there were two types of lures: cross-town lures and same-town lures. Cross-town errors reflected a failure to detect the association between species classes and the towns. Same-town errors indicated that although children may have correctly linked the species classes to their towns, they failed to detect the recurring association between a specific species and its location within that town. We tested whether the proportion of these two error types varied as a function of age. For each task, we conducted a linear regression with the proportion of selected lures as the dependent variable and with age (in months) and error type (same-town versus cross-town) as predictors (see Fig. 4).

In all three tasks, children made fewer errors overall with increasing age (New Animals: $\beta = -0.07$, $SE = 0.009$, $t(238) = -7.79$, $p < 0.001$, 95% CI [−0.083, −0.050]; New Places: $\beta = -0.08$, $SE = 0.008$, $p < 0.001$, 95% CI [−0.101, −0.068]; Baby Animals: $\beta = -0.097$, $SE = 0.008$, $t(238) = -8.45$, $p < 0.001$), 95% CI [−0.082, −0.050]. They were also more likely to make same-town errors than cross-town errors (New Animals: $\beta = -0.02$, $SE = 0.01$, $t(238) = -1.98$, $p = 0.049$, 95% CI [−0.048, −0.0001]; New Places: $\beta = -0.04$, $SE = 0.01$, $t(238) = -3.46$, $p < 0.001$, 95% CI [−0.065, −0.017], Baby Animals: $\beta = -0.03$, $SE = 0.01$, $t(238) = 2.87$, $p = 0.004$, 95% CI [0.01, 0.05]). There was no statistically significant interaction between age and error type for any of the tasks (New Animals: $\beta = 0.002$, $SE = 0.01$, $t(238) = 0.14$, $p = 0.885$, 95% CI [−0.022, 0.026]; New Places: $\beta = 0.02$, $SE = 0.01$, $t(238) = 1.77$, $p = 0.077$, 95% CI [−0.002, 0.045]; Baby Animals: $\beta = -0.02$, $SE = 0.01$,

$t(238) = -1.91$, $p = 0.057$, 95% CI [−0.04, 0.0007]). These results suggest that when children made generalization errors, their mistakes often reflected intact higher-level regularity knowledge of species class-town associations.

## Age-related improvements in memory specificity

To examine the age-related differences in memory specificity, we conducted two linear regression models, one for each index of memory specificity: displacement error and regional accuracy. Note that these two measures were strongly correlated with each other, $\beta = -1.17$, $SE = 0.01$, $t(116) = -84.09$, $p < 0.001$, 95% CI [−1.20, −1.15] (see Fig. S4). Age and verbal intelligence were entered as predictors in both models. As expected, age was negatively correlated with displacement errors, $\beta = -0.007$, $SE = 0.001$, $t(115) = -9.69$, $p < 0.001$, Cohen's $f^2 = 1.07$, 95% CI [−0.008, −0.005] (see Fig. 5A, left). Older children showed greater memory precision in recalling the exact positions of the animals. Similarly, regional accuracy was positively related to age, $\beta = 0.008$, $SE = 0.001$, $t(118) = 9.98$, $p < 0.001$, Cohen's $f^2 = 1.07$, 95% CI [0.006, 0.009] (see Fig. 5A, right), indicating that older children were more likely to place animals in the correct place and town. Trial-level displacement error and regional accuracy are presented in Fig. 5B.

## Contingency of generalization on memory specificity

A central question in this research was whether generalization is contingent on memory specificity, and whether this contingency varies as a function of age during early to middle childhood. To address this, we conducted a series of generalized mixed-effects models using trial-

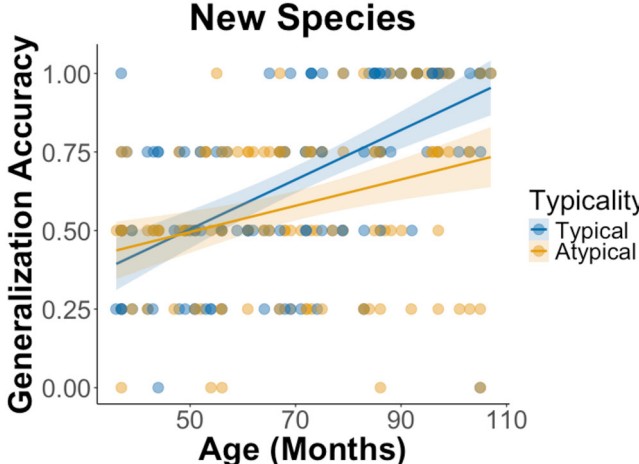

**Fig. 3 | New species generalization accuracy.** Scatterplot of generalization accuracy in the New Species task across age, separated by typical (shown in blue) versus atypical species (shown in yellow). The density of the color indicates the number of participants at a given accuracy level—darker areas reflect more overlapping data points.

level data for each generalization task separately. In each model, we included an age x displacement error interaction term, along with verbal intelligence as fixed effects, and allowed the slope of displacement error to vary by participants. Note that the results remained consistent when regional accuracy was used in place of displacement error as an index for memory specificity (see Fig. S5, Supplementary Note 3). The models for the New Animals, New Places, and Baby Animals tasks were specified as: glmer(generalization accuracy ~ age * displacement error + verbal intelligence + (1 + displacement error | participant)).

For the New Animals task, generalization accuracy was associated with age ($\beta = 1.26$, SE = 0.21, $z = 5.87$, $p < 0.001$, 95% CI [0.84, 1.68]), displacement error ($\beta = -1.38$, SE = 0.21, $z = -6.59$, $p < 0.001$, 95% CI [−1.79, −0.97]), verbal intelligence ($\beta = 0.30$, SE = 0.15, $z = 2.06$, $p = 0.039$, 95% CI [0.01, 0.59]), and an age x displacement error interaction ($\beta = -0.63$, SE = 0.16, $z = -3.97$, $p < 0.001$, 95% CI [−0.94, −0.32]). These results indicate that children's ability to generalize that a new animal exemplar would go to the place as its conspecifics depended on the precision of their memories for the learned animal-location pairs. Further, this dependency varied with age: older children were more likely than younger children to apply learned associations accurately when memory precision was high (see Fig. 6, top left). To further probe this interaction, we examined whether the generalization-specificity contingency is detectable among preschool-aged children (ages 3–5). Indeed, even in this younger subgroup, generalization accuracy was associated with memory precision, $\beta = -1.10$, SE = 0.16, $z = -7.05$, $p < 0.001$, 95% CI [−1.41, −0.80].

We observed similar results for the New Places task. Generalization accuracy was associated with age, $\beta = 1.13$, SE = 0.19, $z = 5.81$, $p < 0.001$, 95% CI [0.75, 1.51], and displacement error, $\beta = -1.76$, SE = 0.20, $z = -8.87$, $p < 0.001$, 95% CI [−2.15, −1.37]. There was no statistically significant effect of verbal intelligence, $\beta = 0.26$, SE = 0.14, $z = 1.90$, $p = 0.058$, 95% CI [−0.01, 0.52]. Again, there was an age x displacement error interaction, $\beta = -0.86$, SE = 0.16, $z = -5.49$, $p < 0.001$, 95% CI [−1.16, −0.55]. As with the New Animals task, children's ability to generalize that a studied animal would go to a similar place was influenced by their memory precision for the animals' placements. This contingency increases with age (see Fig. 6, top right). Nonetheless, among younger children (ages 3–5), this contingency was also significant, $\beta = -1.26$, SE = 0.16, $z = -7.71$, $p < 0.001$, 95% CI [−1.58, −0.94].

In the Baby Animals task, we found the same pattern of results. Generalization accuracy was associated with age ($\beta = 1.64$, SE = 0.20, $z = 7.51$, $p < 0.001$, 95% CI [1.21, 2.06]), and displacement error ($\beta = -1.45$, SE = 0.20, $z = -7.11$, $p < 0.001$, 95% CI [−1.85, −1.05]), but not with verbal intelligence ($\beta = 0.23$, SE = 0.13, $z = 1.79$, $p = 0.074$, 95% CI [−0.02, 0.48]). Again there was an age x displacement error interaction ($\beta = -0.76$, SE = 0.16, $z = -4.70$, $p < 0.001$, 95% CI [−1.07, −0.44]). These findings suggest that children's ability to generalize that an offspring would go to the same place as its same-species adult members depended on children's memory precision for the locations of those adult animals. This dependency intensified with age (see Fig. 6, bottom left). Again, the contingency was found among preschoolers, $\beta = -1.05$, SE = 0.17, $z = -6.03$, $p < 0.001$, 95% CI [−1.39, −0.71].

For the New Species task, we included species-class category knowledge as an additional fixed effect, given its potential relevance to the abstraction required in this task. In this model, displacement error was calculated as an average error across all same-class species (e.g., mammals: horses, pigs, cows, cats) that corresponded to a given test trial (e.g., generalizing that a deer would go to Rubyville). The model was specified as: glmer(generalization accuracy ~ age * displacement error + verbal intelligence + species class category knowledge + (1 + displacement error | participant)). There was no statistically significant effect of memory precision on generalization accuracy, $\beta = -0.05$, SE = 0.16, $z = -0.29$, $p = 0.772$, 95% CI [−0.35, 0.26]. Only age emerged as a significant predictor, $\beta = 0.59$, SE = 0.14, $z = 4.33$, $p < 0.001$, 95% CI [0.33, 0.86] (see Fig. 6, bottom right). To evaluate evidence for the null hypothesis that memory precision was unrelated to New Species generalization accuracy, we computed a Bayes Factor ($BF_{01}$) using the Bayesian information criterion (BIC) approximation[35,36]. We compared the full model that included memory precision (specified above) to a reduced model without it. This method is a frequentist approximation. It assumes flat priors and relies on maximum likelihood estimation from the mixed models. It does not involve Markov Chain Monte Carlo sampling.

The reduced model was specified as: glmer(generalization accuracy ~ age + verbal intelligence + species class category knowledge + (1| participant))

Bayes Factor (BF01) was calculated with the following formula:

$$BF_{01} = \exp\left[1/2 \times \left(BIC_{(reduced\ model)} - BIC_{(full\ model)}\right)\right]$$

The resulting Bayes Factor was $BF_{01} = 8.99 \times 10^{12}$, providing strong evidence in favor of the reduced model. This result suggests that generalizing to entirely new species that are members of the same broader species class is untethered to the precision of memory for specific episodes.

## Unpacking generalization-specificity contingencies

Next, we unpacked the findings that, in the New Animals, New Places, and Baby Animals generalization tasks, the contingency between generalization on memory specificity increases with age. Specifically, we asked *why* this association is weaker in younger children compared to older ones. There were two sources of mismatch between generalization and memory specificity: (1) successful generalization despite unreliable memory specificity, and (2) failed generalization despite reliable memory specificity. In the first case, children accurately generalized when their animal-place associative memories were not entirely reliable. For example, a child may correctly infer that a new horse would go to the castle, even when they had not remembered that the three learned horses were originally placed there. In the second case, children failed to generalize, even though their memory for the original animal-place associations was accurate. That is, a child may have remembered that all three learned horses had gone to the castle, but failed to apply that pattern to a new horse, a baby horse, or a new castle.

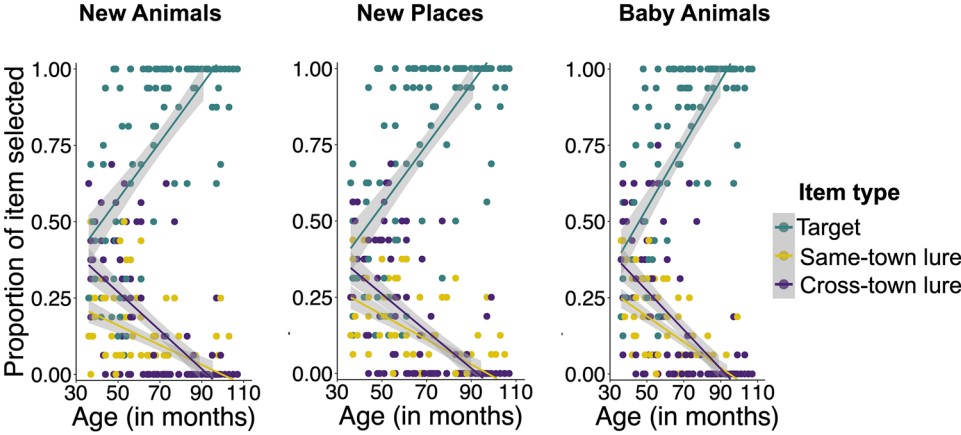

**Fig. 4 | Age patterns in generalization accuracy and error types.** Age patterns in generalization accuracy (target selection shown in teal) and the two error types (same-town lure selection shown in yellow, and cross-town lure selection shown in purple) separated by generalization tasks. Scatterplots of age (measured in months, shown on the x-axes) and proportion of item type selected (shown on the y-axes) in the New Animals (left panel), New Places (middle panel), and Baby Animals (right panel) tasks. Shaded ribbons represent ±1 standard error around the linear regression line for each item type.

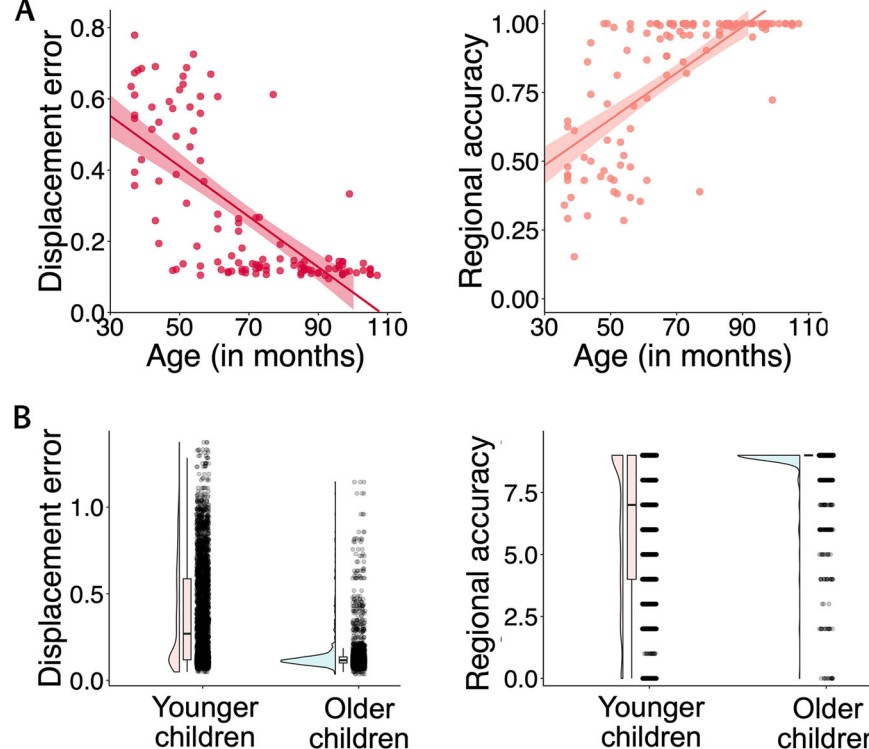

**Fig. 5 | Displacement error and regional accuracy. A** Scatterplots showing participant-level displacement error (left, shown in red) or regional accuracy (right, shown in peach) on the y-axes, plotted against age (measured in months) on the x-axes. Every dot represents a participant. Shaded ribbons represent ±1 standard error around the linear regression lines. **B** Trial-level distributions of displacement errors (left) and regional accuracy (right) shown on the y-axes against age (measured in months) on the x-axes (younger children shown in pink, older children shown in teal). Each dot represents a single trial in the animal-location recall task. The density of the color indicates the number of participants at a given accuracy level−darker areas reflect more overlapping data points.

We defined reliable regional specificity as achieving a perfect score (9 points) across the three animals of a given species (i.e., three regional scores of 3), and unreliable specificity as a sum less than 9. We then calculated the proportion of trials for each child that fell into each mismatch category (successful generalization despite unreliable memory, failed generalization despite reliable memory) for the New Animals, New Places, and Baby Animals tasks separately (see Fig. 7A).

We then compared the proportion of trials between the two mismatching types: accurate generalization despite unreliable animal-place memory, and inaccurate generalization despite reliable animal-place memory, and tested whether they interacted with age. To do this, we conducted separate simple linear models predicting the proportion of trials from mismatch type, age (in months), and their interaction, for the New Animals, New Places, and Baby Animals tasks. The model specification was as follows:

$$\text{lm(proportion of trials} \sim \text{mismatch type*age)}$$

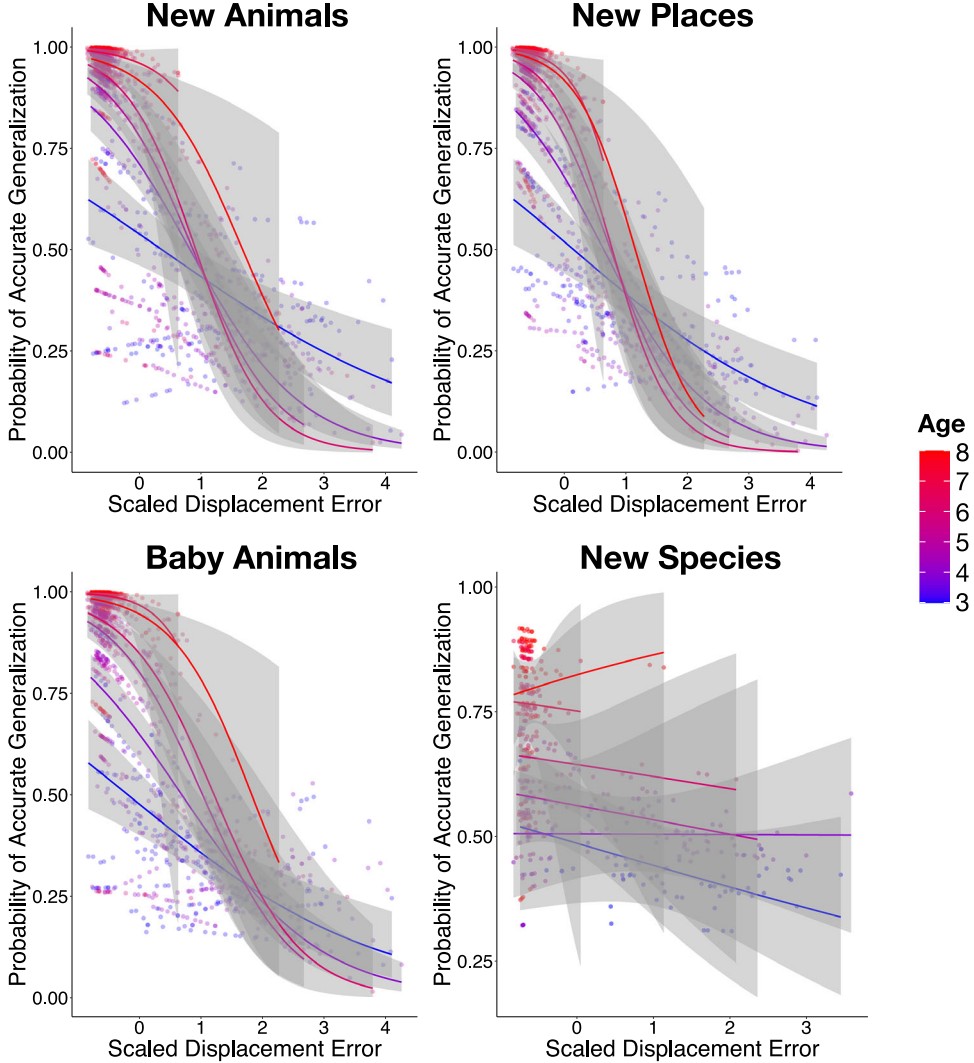

**Fig. 6 | Distributions of generalization accuracy by displacement error.** Distributions of the estimated probability of generalization accuracy on the trial-by-trial basis (*y*-axes) by an interaction between displacement error (scaled, *x*-axes) and age in the New Animals (*n* = 1853), New Places (*n* = 1853), Baby Animals (*n* = 1853), and New Species (*n* = 906) tasks. Each dot denotes an individual trial. The colored lines represent predicted generalization accuracy from binomial mixed-effect models. Color gradients ranging from blue (younger) to red (older) represent children's age in months. The density of the color indicates the number of participants at a given accuracy level–darker areas reflect more overlapping data points. Shaded ribbons represent ±1 standard error per age group. Note that age (measured in months) was treated as a continuous variable in each model and is only separated into age groups for visualization purposes.

For the New Animals generalization task, children were significantly more likely to show accurate generalization with imperfect memory specificity than the reserve mismatch case, $\beta$ = 0.10, *SE* = 0.01, *t*(232) = 7.87, *p* < 0.001, 95% CI [0.078, 0.130]. This difference interacted with age, $\beta$ = −0.10, *SE* = 0.01, *t*(232) = −7.58, *p* < 0.001, 95% CI [−0.128, −0.075], such that the mismatch difference was more pronounced at younger ages and diminished by age 7 (see Fig. 7B). The same pattern emerged in the New Places task. There was a main effect of mismatch type: $\beta$ = 0.09, *SE* = 0.01, *t*(232) = 7.38, *p* < 0.001, 95% CI [0.069, 0.119]. Age interacted with mismatch type, $\beta$ = −0.09, *SE* = 0.01, *t*(232) = −7.07, *p* < 0.001, 95% CI [−0.115, −0.065]. Similarly, for the Baby Animals task, there was a main effect of mismatch type: $\beta$ = 0.09, *SE* = 0.01, *t*(232) = 7.33, *p* < 0.001, 95% CI [0.069, 0.120], which interacted with age, $\beta$ = −0.08, *SE* = 0.01, *t*(232) = −6.03, *p* < 0.001, 95% CI [−0.104, −0.053].

These findings suggest that the weaker association between generalization and memory specificity observed in younger children was primarily driven by cases where children generalized accurately despite having unreliable memory for specific episodes. Older children (ages 7–8) frequently showed both accurate generalization and

memory specificity, which limited our ability to detect mismatches at those ages due to some ceiling effects. Nevertheless, the clear pattern among children ages 3–6 showed that accurate generalization paired with unreliable memory became increasingly prevalent with decreasing age.

## Generalization contingency on memory precision beyond accuracy

Making generalizations about a new animal, a new place, and a baby animal presumably required associative memories of the animal species with their respective place. However, is generalization on these tasks contingent on memory precision above and beyond simply having accurate animal-location associations? We addressed this question in two ways: first, we tested whether generalization accuracy was associated with displacement error even when regional specificity was consistently accurate for a given animal species. That is, when children correctly placed all three exemplars of a species within its respective region, did the precision for the recalled position within that region still impact generalization? Second, we tested whether generalization accuracy was associated with displacement error when

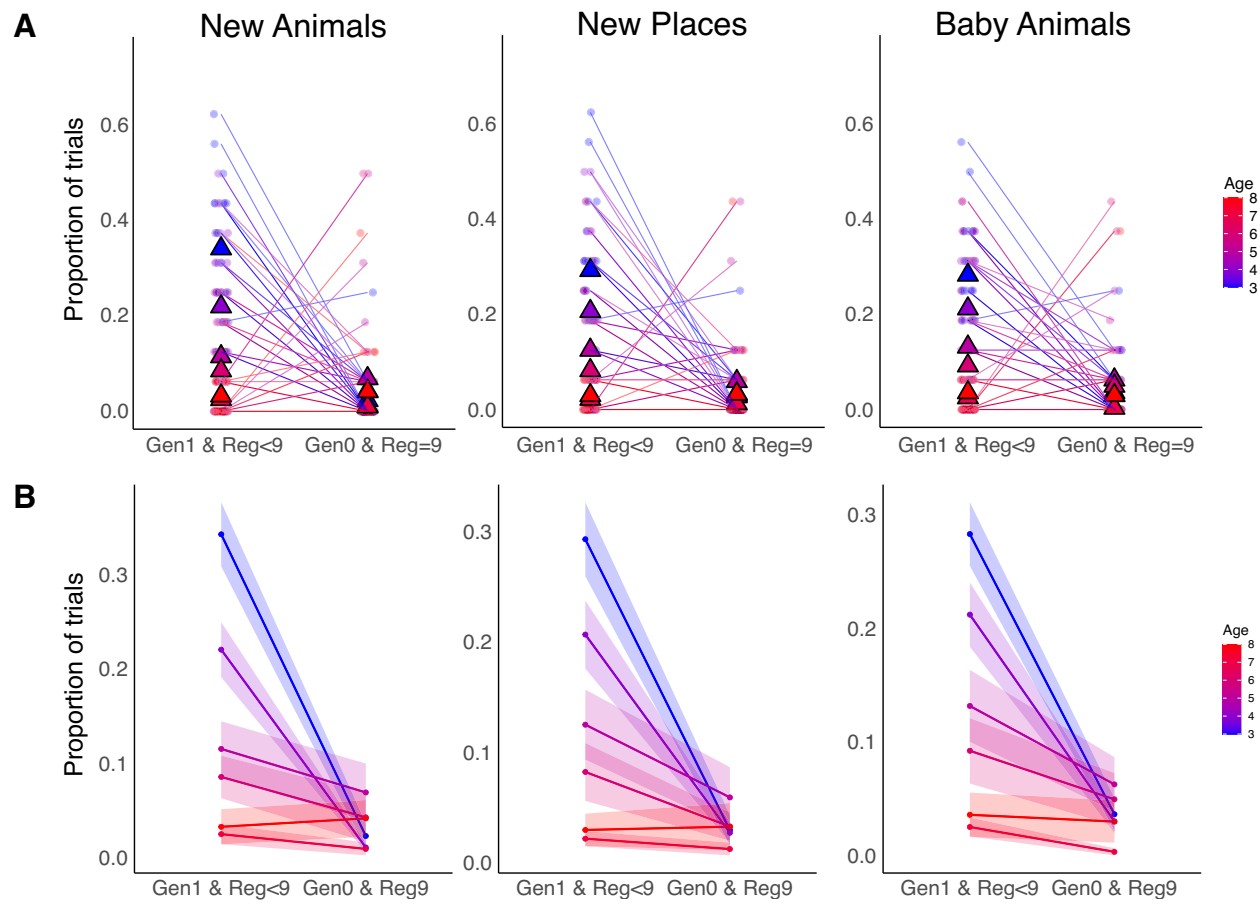

**Fig. 7 | Distributions of Generalization Mismatch Trial Proportions.** Distributions of the proportion of trials that fall into two types of generalization-specificity mismatch: (1) accurate generalization with imperfect regional accuracy score (denoted as Gen1 & Reg<9) versus inaccurate generalization with perfect regional accuracy score (denoted as Gen0 & Reg9) in the New Animals, New Places, and Baby Animals tasks. In (**A**), each dot represents an individual participant. The triangles represent group-level means, plotted separately by age. The group-level means are visualized in (**B**) for visualization purposes. Shaded ribbons represent standard error per age group. Note that age is treated as a continuous variable in all models. The color gradient, ranging from blue (younger) to red (older), represents children's age in months. Overlapping lines create areas of higher visual color density, reflecting clusters of participants with similar response patterns. For the full breakdown of all generalization-specificity combinations, see Fig. S7.

regional accuracy was imperfect. In this case, was it being further off from the original position associated with lower generalization performance?

The first set of analyses, therefore, only included trials with perfect regional accuracy. Each generalized mixed-effects model included age, displacement error, and an age x displacement error interaction as fixed effects, with participants as a random effect. We found that when regional memory was intact, there was no statistically significant effect of displacement error on generalization accuracy in any task (New Animals: $\beta = 0.43$, SE = 0.24, $z = 1.76$, $p = 0.078$, 95% CI [−0.048, 0.901], New Places: $\beta = -0.09$, SE = 0.16, $z = -0.59$, $p = 0.552$, 95% CI [−0.397, 0.212], or the Baby Animals task: $\beta = 0.46$, SE = 0.26, $z = 1.76$, $p = 0.079$, 95% CI [−0.053, 0.973]). Similarly, there was no statistically significant interaction between displacement error and age (New Animals, $\beta = 0.24$, SE = 0.20, $z = 1.20$, $p = 0.230$, 95% CI [−0.155, 0.644], New Places, $\beta = -0.14$, SE = 0.14, $z = -0.96$, $p = 0.337$, 95% CI [−0.412, 0.141], Baby Animals, $\beta = 0.39$, SE = 0.20, $z = 1.89$ $p = 0.058$, 95% CI [−0.013, 0.785]) (see Fig. 8). These findings suggest that once children reliably recalled the correct regional locations, memory for the fine-grained spatial details (i.e., precise position within the region) did not come with additional gains on generalization. In short, accurate animal-place associations were sufficient to support new inferences.

The second set of analyses only included trials with imperfect regional accuracy. We conducted generalized mixed models that predicted generalization accuracy, with displacement error, and an age x displacement error interaction term as fixed effects, and with participants as a random effect (see Fig. 9). In all three generalization tasks, displacement error significantly interacted with age (New Animals: $\beta = -0.24$, SE = 0.11, $z = -2.29$, $p = 0.022$, 95% CI [−0.452, −0.035], New Places: $\beta = -0.37$, SE = 0.12, $z = -3.14$, $p = 0.002$, 95% CI [−0.601, −0.139], and Baby Animals: $\beta = -0.24$, SE = 0.10, $z = -2.32$, $p = 0.02$, 95% CI [−0.441, −0.037]). Larger errors were associated with lower generalization accuracy. This association was stronger with increasing age.

These results suggest that when children's memory for animal-place associations was not reliable, their generalization accuracy decreased as a function of how far off their recalled positions were from the original ones. Interestingly, this association strengthened with age. Older children appeared to be more sensitive to graded memory strength in guiding generalizations.

## Discussion

The current study investigated the development of generalization and its dependency on memory specificity in childhood. We examined how children extracted levels of regularities in animal-place associations

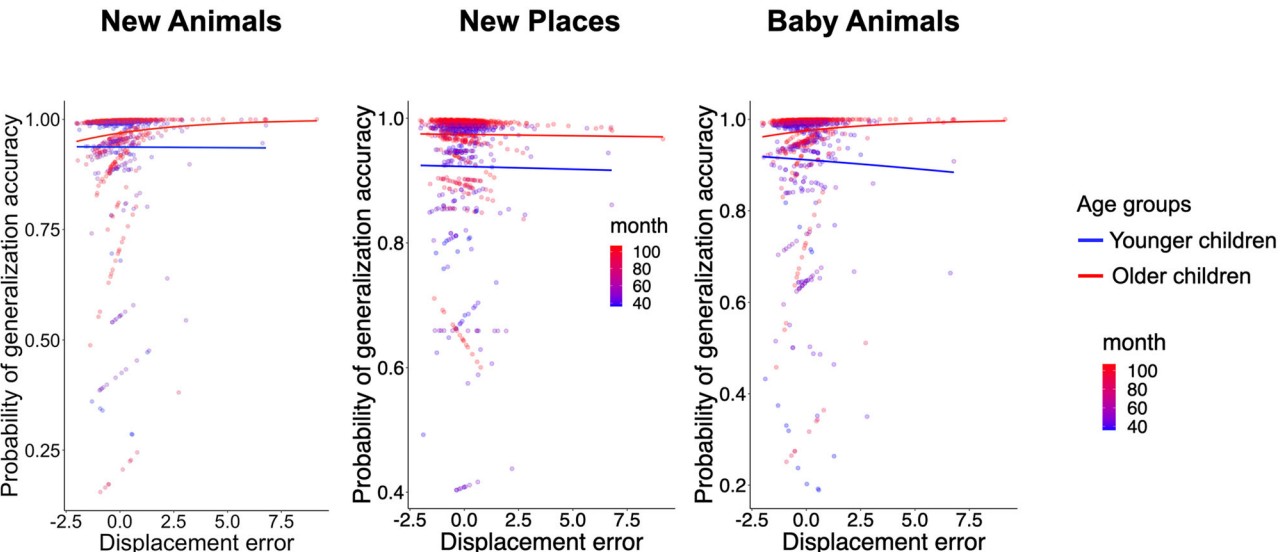

**Fig. 8 | Estimated generalization accuracy by displacement error and age.** Distributions of the estimated probability of generalization accuracy on the trial-by-trial basis (y-axes) by an interaction between displacement error (scaled, x-axes) and age in the New Animals (left, $n = 1210$), New Places (middle, $n = 1210$), and Baby Animals (right, $n = 1210$) tasks. Each dot denotes an individual trial. The color gradient, ranging from blue (younger) to red (older), represents children's age in months. The density of the color reflects the concentration of data points—darker areas reflect more overlapping data points. Note that age was treated as a continuous variable in each model and is only separated into age groups for visualization purposes.

and applied them to new instances across a hierarchy of animal categories. By leveraging the natural taxonomy of animal species, we characterized age-related differences in the ability to extend directly learned associations to new stimuli at varying degrees of proximity to their specific experiences. We also asked whether the contingency between generalization and memory specificity varied across levels of abstraction and with age. There were three key findings. First, generalization across all levels improved significantly with age, and notably, even preschoolers reliably applied their knowledge of recurring animal-place associations to new stimuli across multiple categorical levels. Second, the contingency between generalization and memory specificity depended on both the level of abstraction and the children's age. Lower-level generalization was contingent on memory specificity, whereas higher-level generalization was not. Further, reliance on memory specificity strengthened with age for lower-level generalization. Third, a weaker generalization-specificity association in younger children was primarily driven by their likelihood of generalizing accurately despite having unreliable specific memories, rather than the reverse. This asymmetry diminishes with age.

Across the 3–8-year age range, we observed significant improvements in generalization abilities at all levels, from similar and dissimilar exemplars to higher-order species classes. These age-related differences in generalization could not be attributed to differences in verbal skills. Importantly, even preschoolers (ages 3–5) generalized learned animal-place associations to new exemplars within the same basic-level category, and remarkably, to unseen members of the same superordinate-level category. For instance, after observing several horses go to a castle, children inferred that a new horse should go to the same castle, and that a previously seen horse would go to a new castle exemplar. These findings extend the scope of generalization beyond the agent (e.g., animal or character)[7,24]: children also generalized to new places, demonstrating that knowledge extension is present for each constituent of the learned relationship.

Children could also infer that less perceptually similar exemplars, such as baby animals, would also go to the same castle as adults. This finding suggests that generalization accuracy on the New Animals task is unlikely to reflect poor perceptual discrimination among exemplars[10]. Rather, children's inferences are not necessarily

perceptually bound to their directly learned experiences. These findings align with a prior study showing that 6-year-old children, compared to young adults, were more likely to generalize unobserved properties to perceptually dissimilar exemplars to those seen at learning[37]. Participants were first shown examples of large dogs possessing an attribute (e.g., have sarca inside, or have a strong bite) and later on were asked to judge whether other dogs would also possess these attributes. Compared to young adults, six-year-old children were more willing to generalize these attributes to medium-sized and small dogs. Other studies from this line of research have typically used essential but unobservable anatomical attributes (e.g, this animal has sarca inside). This line of research suggests that children readily generalize essential attributes - beliefs about fundamental properties that define a category – to new members of the same category[38]. However, our results demonstrate that preschoolers can also generalize to non-essential attributes, such as arbitrary associations between locations and animals.

Extending their knowledge a step further, preschoolers could infer that entirely new species, unseen during encoding, would be assigned to the same town as their superordinate category counterparts. Despite this early sensitivity to higher-order regularity, generalization accuracy improved with age over the years of early elementary school, and its relation with age remained after controlling for the effect of Species-Class category knowledge. Thus, age-related differences in the New Species generalization performance were not driven by improved categorical grouping of mammals versus birds.

The age-related improvements in detecting higher-level regularities were also evident in children's error patterns. Across the New Animals, New Places, and Baby Animals tasks, children more often selected same-town lures over cross-town lures, indicating that generalization failures reflected mis-assignments within the correct town rather than a failure to detect higher-level town–species class mappings. This pattern supports the idea that from early to middle childhood, children successfully extracted and applied higher-order regularities, even when their memory for lower-level associations was imperfect.

Children's memory specificity, defined as the precision of their spatial memory for individual animals, improved significantly with age,

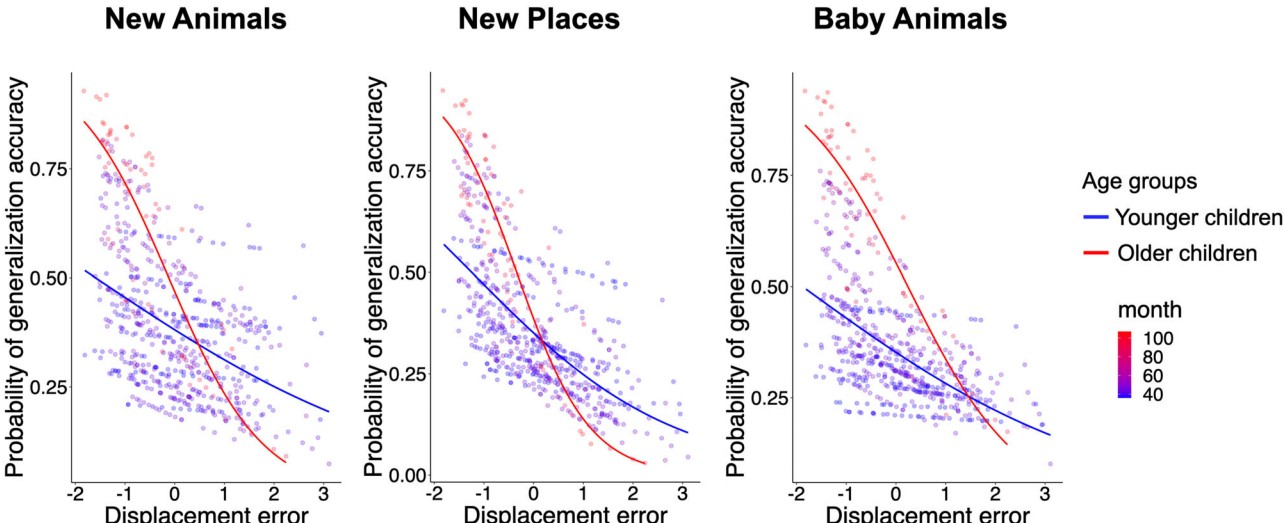

**Fig. 9 | Generalization accuracy by displacement and age when memory is inaccurate.** Distributions of the estimated probability of generalization accuracy on the trial-by-trial basis (*y*-axes) by an interaction between displacement error (scaled, *x*-axes) and age in the New Animals (left), New Places (middle), and Baby Animals (right) tasks when the animal-species specific animal-location associative memory is inaccurate. Each dot represents a single trial. The color gradient from blue (younger) to red (older) represents children's age in months, and the density of the color reflects the concentration of data points—darker regions indicate a greater number of overlapping trials. Age was treated as a continuous variable in each model, but is grouped for visualization purposes.

consistent with prior studies[19,21–23]. A recent literature review found that among the ~30 studies that measured object/event-location memories in children, only two used memory precision - a continuous measure of retrieval error[39]. Our findings support the use of memory precision as a high-resolution index of memory performance, sensitive to both inter- and intra-individual variability.

The second key finding is that the contingency between generalization and memory specificity depended on both abstraction level and age. At higher levels of abstraction, such as generalizing a new species in the same class to a more global space, accuracy was untethered to memory specificity. These findings suggest that higher-level generalizations draw from diffuse or distributed knowledge aggregated across learning episodes. In this case, generalization may emerge from a summary representation, rather than from individual episodic memories.

In contrast, lower-level generalization accuracy was indeed linked to memory specificity. This contingency was evident even among preschoolers, but its strength increased with age. These findings suggested that, as early as age 3, children draw from specific instances to generate inferences about new stimuli, and that there was an age-related increase in the utilization of detailed memory in generating flexible inferences throughout childhood. It is worth noting that memory specificity promotes generalization only up to a point. Once children accurately remembered all exemplars of a species within a region, further memory precision for each exemplar's exact position did not benefit generalization. That is, correct category-place associations were sufficient, and additional fine-grained details did not seem to provide any extra boost.

Conversely, when children's animal-place memories were inaccurate, generalization accuracy declined with greater displacement error. For instance, when children forgot that Peggy the horse had gone to the castle but still vaguely remember that she was somewhere north in Pearl Cove, they would rely on this cue to guide generalizations. Interestingly, this gradient effect strengthened with age, suggesting that older children increasingly leveraged memory traces, albeit imprecise, to inform their generalizations.

Given the limited existing research on multi-level generalization in both children and adults, one speculation is that high-level generalization reflects a form of implicit learning whereby children aggregate

broad patterns across learning episodes without necessarily being aware of or able to recall individual instances. Complex relational structures are often acquired implicitly – unintentionally and without awareness – and such learning mechanisms may support the formation of summary representations over the encoding phase[40–43]. At the neural level, it is possible that lower-level generalizations directly call upon the retrieval of specific episodic memories via hippocampal-dependent mechanisms, either via the trisynaptic pathway in the hippocampus[44], or through the hippocampal big loop[4]. More abstracted representations of the structured knowledge may be built up in the neocortex or via the mono-synaptic pathway within the hippocampus, throughout learning, and are less reliant on the retrieval of specific instances.

The third key finding was that younger children's weaker generalization-specificity link compared to that in school-aged children is driven by their propensity to generalize accurately despite unreliable memories for individual episodes. With age, generalization more closely tracked memory specificity. However, it was not due to preschoolers failing to apply accurate episodic memories. Rather, they were more often able to generalize without strong memory specificity. One note of caution is that this pattern may also be true in older children; near-ceiling performance in both generalization and memory specificity limited our ability to detect it. The opportunity to test the decoupling of generalization and specificity would require a more challenging memory paradigm that yields greater errors in both generalization and memory specificity.

Taken together, our findings clarify the relationship between the two memory functions by showing that, age aside, the nature of their relationship depends on the level of abstraction. It is possible that our multi-level generalization assessment taps into different routes of how individual episodes contribute to generalization[45]. First, generalization can be based on integrating multiple experiences into a summary representation[46]. Second, generalization involves rapidly drawing upon independently stored individual memories[5]. Perhaps the higher-level regularity comes from many instances (24 individual episodes) that spread out over the course of learning, and thus is represented as a summary. In contrast, the lower-level regularity in our paradigm is built from only three instances, leading to a higher likelihood of children's reliance on specific episodic details when generalizing.

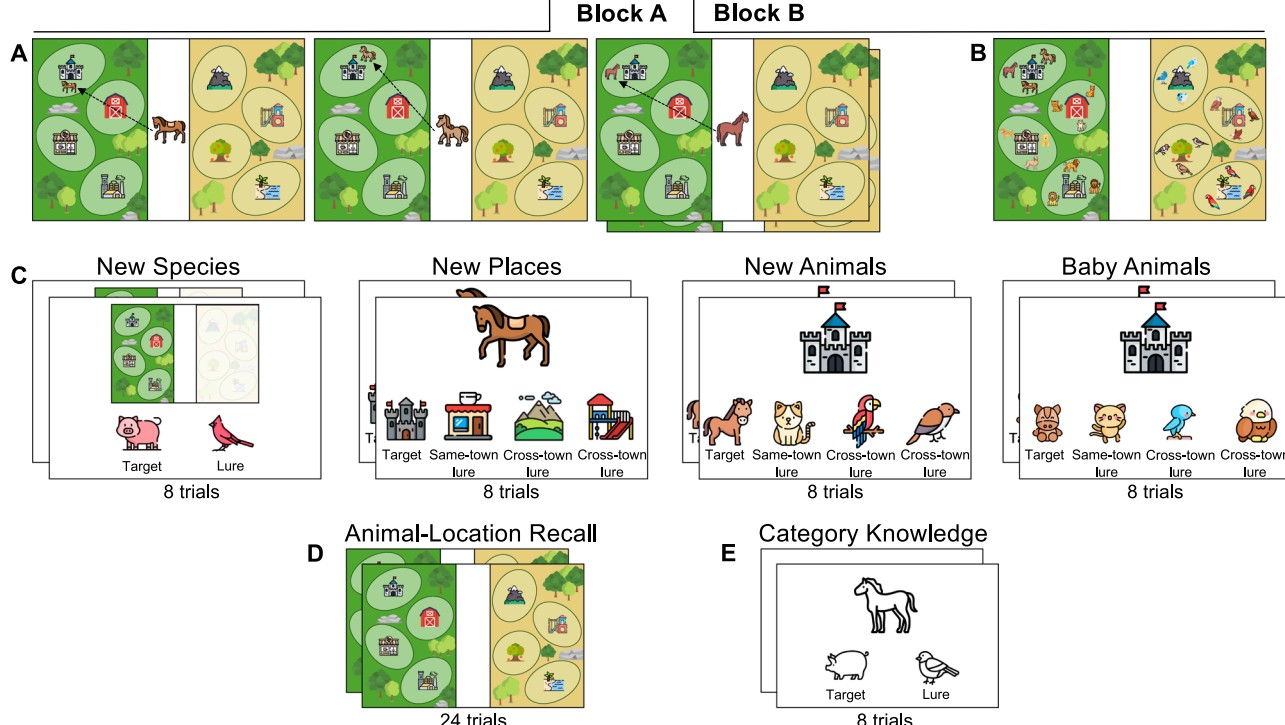

**Fig. 10 | A schematic of the experimental procedure. A** Encoding phase, **B** an overview of all animal-location associations shown sequentially during encoding, which is provided for illustrative purposes only (participants did not see an overview of all animals), **C** the generalization tasks, including the New Species, New Animals, New Places, and Baby Animals tasks, **D** the animal-location recall task, and **E** the species-class category knowledge task. Blocks A and B followed the same procedure but used non-overlapping stimuli. This composite figure is a schematic representation only and does not depict the actual images shown to participants. Created using PowerPoint and icons sourced from Flaticon (www.flaticon.com), distributed under Flaticon's Free License (https://www.flaticon.com/legal). Individual icons were designed by the following creators: ekays.dsgn, Eucalyp, juicy_fish, wanicon, Freepik, max.icons, Mihimihi, Triberion, smalllikeart, Aranagraphics, Flat Icons, Culmbio, Ylivdesign, Graficon, Kroffle, Smashicons, iconixar, yukyik, Pop Vectors, Vector Stall, Valeria, DinosoftLabs, bastian_5, and Irfansusanto20.

In summary, our findings highlight children's impressive generalization ability across multiple categorical levels from early childhood onward. Instead of employing essentialist features that are thought to be crucial for early categorization ability, our paradigm is modeled after real-life situations that require learning arbitrary facts about categories. These findings provide a more nuanced understanding of the relationship between generalization and memory specificity, showing that this relationship is shaped by both the level of abstraction and age of the learner.

## Methods

### Participants

A total of 121 children aged 3–8 years were included in the final sample ($N_{3\text{-year-olds}} = 19$; $N_{4\text{-year-olds}} = 23$; $N_{5\text{-year-olds}} = 19$; $N_{6\text{-year-olds}} = 19$; $N_{7\text{-year-olds}} = 20$; $N_{8\text{-year-olds}} = 21$; 70 female; 59 male; 1 non-binary/prefer not to say; 1 transgender female; $M_{\text{month}} = 70.07$, $SD = 21.21$, range = 36–107). Participants were recruited from Philadelphia and the surrounding suburbs through a Temple University database of families who had expressed interest in participating in research, online advertisements, and community recruitment at events such as farmers' markets. There are no potential self-selection or other biases that may have impacted recruitment or results. The inclusion criteria entailed English proficiency, no history of neurological, learning, or other developmental disorders, nor a history of head trauma, seizures, or brain tumors based on parental reports, and normal or corrected-to-normal vision and hearing. An additional 10 children were recruited but not included in the final sample: one child did not complete the study, and two children did not complete the Peabody Picture Vocabulary Test (PPVT-5)[47], resulting in a sample size of 128 children who completed the full procedure. Seven participants were identified as outliers on the PPVT (see Fig. S2) and, therefore, were excluded from all analyses. This study was approved by the Temple University Institutional Review Board. Informed consent was obtained from the child's parent or guardian. For children 7 years old and above, assent was obtained in addition to parental consent. Participants were compensated with a $25 gift card.

### Materials

All images were selected using the Google Image search engine. For the Home Sweet Home memory task, we created four unique cartoon towns using Photoshop (v22). Each town contained four places (e.g., a café, a castle, a factory, and a barn), with each place occupying one of four oval-shaped sections of the town (see Fig. 10). The area of these oval-shaped sections was kept constant across all places and towns, and the distance between two neighboring places was uniform within each town and across the four towns.

Two of the towns (Rubyville and Pearl Cove) consisted entirely of indoor places (e.g., café, castle), whereas the other two (Emerald City and Topaz Valley) consisted entirely of outdoor places (e.g., apple orchard, playground). Towns were paired into two sets (one per block), and within each pair, one town was indoor-themed and the other outdoor-themed. We selected 16 places based on probable familiarity to children, and chose two cartoon images for each place (e.g., two castle exemplars). Of the 16 places, 8 were outdoor and 8 were indoor.

To create the Home Sweet Home task, we selected 32 animal categories (e.g., horse, cat) and 160 cartoon animal images (four adult exemplars and one baby animal per species). Half of the species were mammals, and the other half were birds. Within each species class of mammals and birds, half of the animals were more typical, whereas the

other half were less typical, selected based on previous typicality norms with children[48] (see Table S1). Each animal exemplar was given a common but distinctive name (e.g., Peggy the horse), drawn from the top 100 male and female baby names in 2016 (https://www.ssa.gov/oact/babynames/). We then assigned each animal category to a specific place (e.g., horse-castle, dog-café), such that all mammals were associated with indoor places and all birds with outdoor places. Further, every exemplar was assigned to a specific positional coordinate within the oval-shaped area of a given place. Two non-overlapping sets of towns and stimuli were constructed. For each set, we generated eight pseudorandomized versions of encoding and test phases such that every animal exemplar appeared in the encoding phase an equal number of times across participants. The eight versions varied in both encoding and test trial order.

For the Category Knowledge task, we selected schematic black-and-white animal images for 24 species from Flaticon (https://www.flaticon.com/). These 24 animal species were identical to those used in the Home Sweet Home memory task.

### Overall procedure

All participants were tested individually. The experiment consisted of three tasks administered in a fixed order across participants: the Home Sweet Home memory task, the Species-class Category Knowledge task, and the Peabody Picture Vocabulary Test Fifth Edition (PPVT-5)[47].

**Home Sweet Home Memory task**. The Home Sweet Home task was divided into two successive encoding-test blocks, each with non-overlapping stimuli. To maximize coverage of the species class, one block featured species with higher typicality, and the other featured species with lower typicality (see Table S1). The order of these two blocks was counterbalanced across participants.

**Encoding**. Children were given a cover story in which animals needed to find homes in two fantasy towns, and their job was to help the animals if they got lost. They were introduced to the two towns (e.g., this is Rubyville and this is Pearl Cove) and were instructed to pay close attention to where each animal went to find its new home. The towns were presented simultaneously, one on the left and the other on the right side of the screen. Each encoding phase consisted of a pre-recorded animation containing 24 trials of animal-location pairings. On each trial, an animal first appeared in the center of the screen for 2 s before moving to a specific position within its assigned place and town (see Fig. 10A). A pre-recorded voice narrated each trial (e.g., Peggy the horse goes to the castle), and a chime sound accompanied the animal's arrival to maintain children's engagement throughout the task. Each encoding trial lasted 14 s, and the animal disappeared at the end of its respective trial.

In an interleaved fashion, children saw that all animals from the same-species class moved to the same town (e.g., mammals to Rubyville with all indoor places; birds to Pearl Cove with all outdoor places). Three different exemplars of each species (e.g., three different horses) were assigned to the same place (e.g., castle), but each occupied a distinct position within the oval-shaped area of that place. All animal positions were equidistant from the center of their assigned place and from each other within the oval section (see Fig. 10B). The encoding trial order was pseudorandomized such that (1) no two exemplars of the same species appeared in direct succession, and (2) no more than two trials from the same animal class occurred consecutively.

**Testing phase**. Immediately after encoding, the test phase included four generalization tasks (see Fig. 10C), followed by the animal-location recall task. The four generalization tasks – New Animals, New Places, Baby Animals, and New Species – all preceded the animal-location recall task. However, their order was pseudorandomized across participants, with the constraint that the New Species task was

always administered either first or last. The New Animals generalization task and New Places generalization task were randomized in order and always occurred consecutively, and were always followed by the Baby Animals task. Each child received one of the eight pseudorandomized task versions that corresponded to their encoding version.

**New Animals Generalization task**. This task assessed whether children could generalize that a new exemplar of a species (e.g., a new horse) would move to the same place as its conspecifics. It required children to acquire and deploy animal-place association patterns to new and perceptually similar animal exemplars. There were eight self-paced 4-alternative forced-choice (AFC) test trials, one per animal species in each block. On each trial, children saw a place previously seen at encoding (e.g., the castle) at the top of the screen and four animal options at the bottom. They were asked to choose the animal that would move to that place to be with their family. All options were new exemplars of the species seen at encoding. The target was the new exemplar of the species that was previously paired with the place. The same-town lure was a new exemplar of a species that had appeared in a different place but in the same town as the target (e.g., a new cat). The two cross-town lures were new exemplars of species from the other town (e.g., a new sparrow, a new cardinal). The positions of the target and lures were counterbalanced across participants. Across the eight versions and within each version, each animal served as the target or a lure an equal number of times.

**New Places Generalization task**. This task assessed whether children accurately generalized that a studied animal (e.g., Peggy the horse) would move to a new exemplar of the place where it and its species had previously settled (e.g., another castle). Parallel to the New Animals generalization, this task required children to acquire and deploy animal-place association patterns, but this time to new and similar places. There were eight self-paced 4AFC test trials, one per place in each block. On each trial, a studied animal (e.g., Peggy the horse) appeared at the top of the screen, with four place options below. Children were told that some animals had gotten lost and needed help finding their family by choosing the place the animal would most likely go. The target was a new exemplar of the place previously paired with that animal species (e.g., a new castle). The same-town lure was a new exemplar of a different place in the same town (e.g., a new café), and the two cross-town lures were new exemplars of places from the other town (e.g., a new apple orchard and a new playground). The positions of the target and lures were counterbalanced across the trials. As with the New Animals task, each place served equally often as a target or lure across all versions and within each version.

**Baby Animals Generalization task**. This task assessed whether children could generalize that a baby animal would move to the same place as its adult conspecifics. Baby animals are more perceptually distinct from adult animals. Thus, this task required the deployment of animal-place patterns to a new instance that was less perceptually bound to their directly learned experiences than the New Animals and New Places tasks. There were eight self-paced 4AFC test trials, one per animal species in each block. On every trial, a studied place (e.g., the castle) appeared at the top of the screen, and four animal options appeared at the bottom. All options were baby exemplars of the animal species seen at encoding. Children were asked to choose the baby animal that would move to that place to be with their family. The target (e.g., a baby horse) was the baby of the species previously paired with that place. The same-town lure was a baby of a species assigned a different place in the same town (e.g., a baby cat), and the two cross-town lures were baby animals of species from the other town (e.g., a baby sparrow, a baby cardinal). Target and lure positions were counterbalanced across the trials. Importantly, the trial order and the 4AFC combinations differed from those in the New Animals generalization

task. Each animal served as the target or as a lure an equal number of times across versions and within each version.

**New Species Generalization task.** This task examined whether children could extract the broader rule that all members of a species class (mammals or birds) were assigned to the same town, and apply that knowledge to unstudied animal species. This high-level generalization, therefore, relied on the aggregated knowledge built from the learning phase. There were four self-paced 2AFC test trials (two per species class) with unique species. On each trial, children were shown two towns, one highlighted and the other faded, presented alongside two unstudied animal species, one mammal and one bird (e.g., a pig and a crow). They were asked to choose which animal would want to move to the highlighted town. The target was the animal from the same-species class assigned to that town during encoding. The lure was an animal from the opposite town.

**Animal-Location Recall task.** This task assessed children's memory for the precise location where each animal had been placed during encoding. There were 24 self-paced test trials in each block. On each trial, the two towns were shown in the same layout as during encoding, and a studied animal appeared in the center of the screen (see Fig. 10D). Children were asked to drag and drop the animal to the exact position where it had previously been located. They could place the animal anywhere on the screen, including outside of the town, or place boundaries if they wished. To minimize motor-related variability, this task was administered on a laptop with an external mouse. If a child had difficulty using the mouse, the experimenter instructed them to touch the desired location on the screen, holding their finger in place until the animal was moved accordingly by the experimenter. The experimenter then confirmed the selected location with the child before proceeding to the next trial to ensure the placement reflected the child's intention before proceeding to the next trial. The test trial order was pseudorandomized such that no more than two animals from the same-species class or species appeared in direct succession. The test trial order also differed from the encoding order.

**Species-Class Category Knowledge task.** The species-class category knowledge task assessed children's understanding of mammal versus bird classifications, using the same set of species from the Home Sweet Home memory task. To ensure that the stimuli were distinct from those in the other tasks, we used child-friendly schematic images of mammals and birds that were devoid of color and stylistic detail. This non-overlapping stimulus set reduces the influence of the Home Sweet Home task on this task. Children were told a cover story: Zuga, an alien from outer space, had discovered various animals on Earth. On each trial, children saw an animal that Zuga had found (e.g., a horse) at the top of the screen, and two animals - one mammal and one bird (a pig and a bluebird), at the bottom. They were asked to decide which of the two animals was more similar to the one Zuga found. This task consisted of eight self-paced 2AFC trials, 4 per species class, and all participants completed the same version (see Fig. 10E).

**Verbal Intelligence.** Verbal intelligence was measured using the Peabody Picture Vocabulary Test Fifth Edition (PPVT-5)[47], a commonly used measure of receptive vocabulary for individuals aged 2 to adults 90+ years old. On each trial, the experimenter presented four images for a given word (e.g., foot), and the child was asked to select the image that best represented the word's meaning. The average completion time was 10–15 min. Raw scores were standardized based on age for all analyses.

**Statistical analyses**
All statistical analyses were conducted in R 4.1.2 using RStudio 2022.07.2 Build 576[49,50]. The dependent variables in each generalization task included accuracy (proportion of target selection across all test trials) and two different types of errors: same-town and cross-town errors (proportion of each lure type selection across all test trials).

We estimated memory specificity in the Animal-Location recall task with two indices: a displacement error and a regional accuracy, calculated for each of the 24 test trials. Displacement error was defined as the Euclidean distance between the animal's true coordinate and the participant's recalled placement. This serves as a sensitive measure of memory precision. However, displacement error alone is agnostic to whether animals were placed within or outside the boundaries of a given place. Therefore, we also calculated regional accuracy, which accounted for both town and place boundaries. Participants received a score based on the following criteria: 3 points for placing the animal in the correct town and within the boundary of the correct place, 2 points for placing the animal in the correct town but in the boundary of an incorrect place, 1 point for placing the animal in the correct town, but outside the boundary of any place, and 0 points for placing the animal in the wrong town, whether within a place, outside of a place, or outside of either town altogether. Experimenters recorded regional scores manually on paper for each trial.

Due to a technical error in PsychoPy, data from five children were not recorded for one block of the animal-location recall task. In addition, 14 participants had missing data (ranging from 1 to 4 trials per participant), which occurred at random due to the experimenter or child skipping a trial unintentionally, or technical errors during task administration.

**Reporting summary**
Further information on research design is available in the Nature Portfolio Reporting Summary linked to this article.

## Data availability
The processed data generated in this study have been deposited in the Open Science Framework repository under https://doi.org/10.17605/OSF.IO/2GAHZ [https://doi.org/10.17605/OSF.IO/2GAHZ][51]. All data are available and are not restricted.

## Code availability
De-identified data, analysis code, and experimental materials have been made publicly available through the Open Science Framework at [https://doi.org/10.17605/OSF.IO/2GAHZ][51].

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

## Acknowledgements

We would like to thank Kara Storjohann, Linh Nguyen, Kate Hill, and Giovanna Arantes de Oliveira Campos for their help with stimulus

development and/or with data collection. This work was supported, in part, by the German Research Foundation (DFG; NG 191/2-1) and a Research Fellowship by The Jacobs Foundation (2021-1417-99) to C.T.N., and by R01HD099165 from the National Institutes of Health to N.S.N. The content is solely the responsibility of the authors and does not necessarily represent the official views of the National Institute of Health.

## Author contributions

All authors developed the research questions and the design of experiments. Data were collected by S.K. and others (see "Acknowledgements" section). Data analyses and results interpretation were led by S.K., under the supervision of N.S.N. and C.T.N. All authors drafted the manuscript and approved the final version of the manuscript for submission.

## Funding

## Competing interests

The authors declare no competing interests.
