## [Transparent Peer Review file · Nature Communications]

The dependence of children's generalization on episodic memory varies with age and level of abstraction

Corresponding Author: Ms Sabrina Karjack

Version 0:

Reviewer comments:

Reviewer #1

(Remarks to the Author)

In this manuscript, Karjack et al. examine the relation between memory specificity and generalization in a group of school-aged children. Children watched videos of animals going home to different locations. Each location was nested within a place, which was nested within a town. Each animal was nested within a species, which was nested within a species class (mammal & bird). All places were home to one species, and all towns were home to one species class, in theory permitting participants to generalize a.) the town that would be home to new animals, b.) the type of place that would be home to a familiar animal, and c.) the place that would be home to a new exemplar from a specific species. The researchers tested children's generalization as well as their specific memory for the animal locations. They also examined and controlled for children's verbal intelligence and category knowledge. In line with their prior findings, the researchers found that generalization depended on specific memories, but the strength of the association between mnemonic specificity and generalization increased with age. The novel contribution of this particular study was examining generalization at two levels (e.g., to places and to towns). The higher level generalization was not dependent on mnemonic specificity, according to the specific analyses the researchers ran.

This manuscript has many strengths, but it is largely a conceptual replication of two prior studies by some of the same researchers (Ngo et al., 2021 & Buchberger et al., 2024). In terms of strengths — the manuscript was well-written and the task was well-designed and carefully controlled. However, the strengths of the task design replicate the strengths of the design of the task the authors used in prior studies. It was not clear to me that the addition of the higher level of generalization represents a significant conceptual advance above and beyond the authors' prior work, particularly because the hypotheses for high-level generalization were unclear. In particular, the authors ask: "First, can children generalize through extracting regularities across multiple instances and does this ability significantly improve with age? Second, is children's successful generalization dependent on memory specificity of the individual instances?" But both of these questions were already (beautifully!) addressed in Ngo et al. 2021.

Here, the authors additionally ask whether the relation between mnemonic specificity and generalization depends on the level of generalization. The authors note that the development of superordinate categorization emerges around 3 to 4 years. But the youngest participants in their samples were age 4. So it is not clear why the later emergence of superordinate vs. basic-level categorization should affect behavior in the age group they studied. In addition (and I describe this in more detail below), because of the structure of the task, higher-level generalization does not necessarily require the same number of specific memories or the same level of mnemonic precision that lower-level generalization requires. Thus, the lack of relation between memory specificity and high-level generalization seems like a potential feature of the design of this task, and not a general property of memory.

Related to this latter point, I also have some concerns and suggestions about aspects of the analytic approach and interpretation. I lay these out in more detail below. I believe that these concerns would be addressable in a revision; however, while I think addressing them would strengthen the manuscript, I still do not believe the conclusions the authors will be able to draw will be different from what they have already shown in two prior, published studies.

1. Throughout the manuscript, it would be helpful if the authors more explicitly stated what information / memories / knowledge would actually be required for each test of generalization, and analyze the data through this lens. As two examples: I was somewhat confused as to why the authors analyzed the relation between displacement accuracy and generalization in the New Animals task, rather than analyzing the relation between place memory accuracy and

generalization. The authors do eventually run this analysis, and, unsurprisingly, find that displacement accuracy does not predict generalization above and beyond place memory accuracy. This is not surprising: To successfully generalize in this case, one only needs to know the place of the other animals, not their precise locations. Thus, it was unclear why displacement accuracy was the main focus.

2. Similarly, I was confused by the dichotomization of memory specificity in the analyses examining why the relation between specificity and generalization increases with age. In particular, the authors report the finding that children are often able to successfully generalize despite having imperfect memories for the animal-place associations. This is potentially interesting, but again, given basic rules of logic, it doesn't seem like perfect memory is required for generalization. Instead, it seems like successful generalization would be possible in all these cases if participants remembered that two out of three of each animal lived in a specific place. In other words, the authors should clarify whether young children can generalize in the complete absence of relevant specific memories, or whether they are simply making inferences from 'noisier' and less accurate (but still accurate enough to be useful) specific memories. In fact, a very interesting follow-up experiment would be to 'train' adults on the basis of children's specific memories and then ask them to perform the generalization tasks. In other words, rather than providing adults with the standard learning phase, the researchers could yoke each adult participant to a child, and teach them their remembered animal locations. In this case, if adults' generalization appears comparable to children, then this would suggest that children are not necessarily relying on different types of memory representations or strategies to perform the generalization tasks, but rather that their less precise memories still permit generalization. The authors conclude that children are still able to generalize based on imprecise memories, but it seems like it's possible that anyone (regardless of age) could similarly generalize from these types of memories, and children just happen to have more imprecise memories. Indeed, from Figure 6, it seems like older children are actually better than younger children at generalizing from imprecise memories (e.g., for New Animals, older children successfully generalized in 59% of cases where they had imperfect memory compared to 38% for younger children). This is actually the comparison that seems most interesting, though it was not tested statistically. I'm not necessarily suggesting the authors run the yoked-adult study, but they should carefully consider their interpretation of the relation between specificity and generalization given that generalization still seems possible even on the basis of fewer accurate specific memories.

3. Relatedly, in addition to avoiding the binary division of perfect vs. imperfect regional memory, the authors should also avoid the arbitrary binary division of age group.

4. Similar to my first comment, it was not clear to me why the authors used displacement error in their analysis of generalization in the New Species task. It is not surprising here that displacement accuracy does not predict generalization, because remembering the specific locations of animals is unnecessary to perform this task successfully. It seems as though to successfully generalize in this task, participants need to simply remember that the majority of mammals came from Town A and the majority of birds came from Town B. Presumably the researchers can compute whether participants remembered this information based on participants' placement of the animals in the location memory test. Again, as stated in my initial evaluation of the manuscript, it is unclear what the lack of relation between specificity and generalization here really tells us about memory development, because highly specific memories do not seem required to perform this generalization task. It seems interesting to determine whether children are performing this generalization task by drawing on imprecise specific memories OR by drawing on some type of aggregate / generalized / incrementally updated representation that is not dependent on individual episodes at the time of retrieval, but it does not seem like the authors have the data to disambiguate these two possibilities here.

5. I am concerned that the authors did not include random participant slopes in any of their models. The inclusion of intercepts only may be inflating the Type I error rate.

Minor:

1. Why was the regional accuracy score manually coded as opposed to derived from the location coordinates of the animal placement?

2. While I eventually understood the authors' intended meaning, it would be helpful to clarify what "accurate generalization in the face of unreliable animal-place associations" means upon first mention. Initially, I was confused about whether 'accurate generalization' here means accurate generalization given incorrect beliefs. (i.e., the participant selected the wrong answer but 'generalized' correctly — they incorrectly remember that most horses live in the house and then correctly extend that incorrect belief to novel horses) or accurate generalization given the true structure of the task.

3. Although it didn't detract from the manuscript, the purpose of the baby animals generalization task was not clear to me.

(Remarks on code availability)

The code is available and clearly commented, but I have not looked at it closely or tried to run it.

Reviewer #2

(Remarks to the Author)

The manuscript by Sabrina Karjack, Nora S. Newcombe, and Chi T. Ngo addresses important questions, namely how children manage to generalize, and how the role of episodic memory in this process changes with age. This topic is both timely and not well understood, making the study a valuable contribution. The study design is rigorous and effectively investigates various levels of abstraction, while also capturing episodic memory performance in children aged 3 to 8 years.

The findings suggest that higher-level generalization in children is not reliant on detailed, specific memories. On the other hand, lower-level generalization shows a dependence on specific, detailed memories, and that this reliance increases with age. While the manuscript is well written and insightful, there are some areas that would benefit from further clarification.

1. In the abstract, the phrase "...characteristics of individuals..." seems to imply individuals as in persons/beings. The authors might have meant individual experiences or episodes. Can the authors please clarify?
2. There are several jargons in the abstract that are used interchangeably without definitions. These are memory specificity/memory precision, and abstraction/generalization/regularities. Revising the use of these terms will increase readability.
3. It would be helpful if the authors could provide the study hypotheses alongside the research questions. Was the study preregistered?
4. Could the authors provide a more detailed breakdown of the ages of the excluded participants? Providing descriptive statistics of the PPVT task (with age) will also be helpful.
5. In the description of the typicality of the task stimuli, it is stated that "Within each animal species, half of the animals were more typical, whereas the other half were less typical." Could the authors clarify what the typicality was based on? Was it determined using previous typicality norms with children (e.g., Posnansky, 1978)?
6. It is unclear from the manuscript whether any correction for multiple comparisons was applied. Given the repeated testing of similar models on related measures (of generalization), should this be taken into consideration?
7. Can the authors clarify how displacement errors were included in the various models in Section 2.1? For the new animals/new places/baby animals models, were displacement errors aggregated within each animal species? For the model of New Species (p. 22), did the authors examine whether displacement errors of same-class species are actually correlated? The reasoning here is that displacement error in this model was treated differently from the other models. Given the central role of the findings in this section to the study, it would be important if the authors could verify to what extent displacement error as a predictor is comparable in the measurement properties across the lower- vs. higher-level generalization tasks.
8. Related to the question above, in Figure 5, it looks like the range of New Species generalization success (the y-axis) is more restricted compared to the other generalization tasks. Could this have contributed to the lack of relationship with displacement error in this task?
9. Did the new species stimuli also differ by typicality? Did this affect performance on New Species generalization?
10. While I appreciate the rationale and potential insights offered by Section 2.2, it is difficult to follow what kind of analysis was done to compare the red and blue circled cells of Figure 6. Specifically, as the denominators of these cells are different, is it meaningful to directly compare them to each other statistically (if this was done)? Some elaboration on the statistical comparison would be helpful.
11. In the discussion, the authors stated, "This set of results suggests that from early to middle childhood, children were able to recognize and apply higher-level patterns between the species classes and towns, even when they failed at the lower-level associations between species and specific places." However, the manuscript does not explain why this might be the case. Could the authors elaborate on what aspects of cognitive or brain development may contribute to the early onset of this ability or the formation of "summary" representation?

(Remarks on code availability)

Reviewer #3

(Remarks to the Author)

The present study assessed generalization across different levels of abstraction and relations with memory specificity in 128 children ages 3-8 years old. Generalization across all levels improved with age. While higher-level generalization was not related to memory specificity, lower-level generalization was such that the relationship between generalization and memory specificity strengthened with age. Younger children seem to be able to generalize despite imprecise episodic memories. This manuscript is clearly written and addresses several gaps in knowledge regarding generalization in children. Below are some comments and questions that seek to clarify the motivations for aspects of the study design, details relating to the results, and interpretations of the findings.

-The hypotheses were not clear - there was not a clear motivation for including each level of abstraction that was tested in this study. Were differences in the relation between memory specificity and generalization at different levels of abstraction expected? What was the rationale for including each of the 4 generalization tasks? These points could have been addressed more clearly in the introduction and methods.

-Related to the previous point, the animal-location recall task also seems to not only be a more difficult task (not forced choice) but also a more difficult motor movement. Are there any data that speak to the youngest children's ability to accurately move the animal to their desired position? This seems important for interpretations regarding memory specificity.

-Several aspects of the results section warrant further clarification. Correlation values for stats in lines 361-364 should be stated to get a sense of the effect sizes. Result statistics assessing the effect of age on new species generalization task (lines 382-384) are missing along with model set up details (was species class category knowledge an additional predictor to PPVT?). What statistical tests are used in assessing generalization-specificity mismatch (Figure 6)?

- Was the effect of age on generalization accuracy tested while considering both species class category knowledge and PPVT for all generalization tasks? Better knowledge of species class category could help participants at the least in the new and baby animals task.

-Based on the intercepts of the lines of fit in Figure 2, the youngest participants appeared to be, on average, above chance on the new animals, new places, and baby animals tasks (where chance is 25%) and below chance on the new species task (where chance is 50%). Could this (or the compressed range/decreased variation in performance for the new species task) have contributed to the lack of age by displacement error effect on this task versus the others, especially given that effects seem to trend in the same direction as the other generalization tests? Some caution in interpretation seems warranted.

-Regarding the generalization-specificity mismatch, could the lack of effect for older children be due to ceiling performance? It looks like for most trials, older children have good specificity, so there are very few trials available to assess potential differences between the two generalization-specificity mismatches. Additionally, conditioning by specificity, it appears that for both younger and older children accurate specificity implies better generalization. Conversely, inaccurate specificity seems to imply worse generalization for younger children while being slightly better for older children (bottom rows in Figure 6). Does this not provide evidence that younger children do indeed rely on memory specificity for generalization contradicting the third conclusion as stated in the discussion section? That is, when memory specificity is worse, younger children generalize worse whereas older children do not. If younger children indeed did not rely on memory specificity relative to older children, their generalization when memory specificity was low should be relatively higher compared to older children which doesn't appear to be the case.

Minor:

- Given the possible reliance on audio cues, it would be useful for a reader to know whether all participants were native English speakers.

- Figure 1B shows an overview of the world state after a block, whereas the text implies that participants were shown this world state 'sequentially' where each animal disappeared after moving to its respective home location. Were the participants ever shown the overview in Figure 1B? I think not, but it would be useful to explicitly state so. If they were shown, potential influence of the displayed complete world state on subsequent performance should be discussed.

- Participants are presented animals with low and high typicality however data appears to be analyzed after combining across both. Are there any meaningful differences in generalization performance between high vs low typicality?

(Remarks on code availability)

An access code was required to view the code

Reviewer #4

(Remarks to the Author)

(Remarks on code availability)

Code link lead to a request for access.

Version 1:

Reviewer comments:

Reviewer #1

(Remarks to the Author)

My previous assessment of this manuscript was that it was largely methodologically sound, but potentially not that novel given how closely related it is to the authors' previous work. I believe the authors have done a nice job of responding to my methods concerns, and I think the new analyses and figures are helpful in understanding the complex relations between memory specificity and generality. I believe this work has been done very well, even more so after the revision. That said, given that generalizing more broadly logically does not require detailed memories, I still do not believe that the additional level of hierarchy necessarily extends our understanding of memory development that much above and beyond existing work.

(Remarks on code availability)

Reviewer #3

(Remarks to the Author)

The authors have largely addressed our comments and questions.

We appreciate the updated analyses conducted to assess the age-related differences in the coupling of memory specificity and generalization performance. In Figure S5, it would be useful to show where the curve and model statistics extend for the cases with 100% memory specificity as well. For example, if older kids' generalization stays as it is at 100% specificity or if younger kids' generalization gets a non-linear boost at 100% specificity, the coupling strength by age effect could go away. Generally the authors should also consider the lack of age by regional accuracy interaction in the new animals task in their conclusion.

(Remarks on code availability)

The code is clear and annotated.

Reviewer #4

(Remarks to the Author)

(Remarks on code availability)

Reviewer #5

(Remarks to the Author)

The authors have successfully addressed most of my comments. However, I believe the Discussion section would benefit from including the information raised in Comment 11, as it touches on potential mechanisms underlying the key findings. Although this aspect is not central to the manuscript's primary goals, its inclusion would provide a more comprehensive interpretation of the results within the context of the broader neuroscientific literature.

Comment 11: In the discussion, the authors stated, "This set of results suggests that from early to middle childhood, children were able to recognize and apply higher-level patterns between the species classes and towns, even when they failed at the lower-level associations between species and specific places." However, the manuscript does not explain why this might be the case. Could the authors elaborate on what aspects of cognitive or brain development may contribute to the early onset of this ability or the formation of "summary" representation?

Given the limited existing research on multi-level generalization in both children and adults, we can only speculate on why high-level generalization appears to be less reliant on memory for individual episodes. One possibility is that high-level generalization reflects a form of implicit learning whereby children aggregate broad patterns across learning episodes without necessarily being aware of or able to recall individual instances. aggregated across the learning phase. Prior work has shown that complex relational structures are often acquired implicitly, that is, unintentionally and without awareness (Cleeremans et al., 1998; Garvert, Dolan, & Tim Behrens, 2017; Reber, 1989, Seger & Augart, 1994). Such learning mechanisms may support the formation of summary representations over time. At the neural level, it is possible that lower-level generalizations directly call upon the retrieval of specific episodic memories via hippocampal-dependent mechanisms, either via the trisynaptic pathway (Shapiro et al., 2017) in the hippocampus, or through the hippocampal big loop (Koster et al., 2019). More abstracted representations of the structured knowledge may be built up in the neocortex or via the mono-synaptic pathway within the hippocampus, throughout learning and are less reliant on the retrieval of specific instances. Cleeremans, A. Destrebecqz, & Boyer, M. (1998). Implicit learning: news from the front. *Trends in Cognitive Sciences*, 2, 406-416.

Garvert, M. M., Dolan, R. J., & Behrens, T. E. J. (2017). A map of abstract relational knowledge in the human hippocampal-entorhinal cortex. *Elife*, 6, e17086.

Reber A. S. (1989). Implicit learning and tacit knowledge. *Journal of Experimental Psychology: General*, 118, 219-235.

Seger, C. A., Augart, C. (1994). Implicit learning. *Psychological Bulletin*, 115, 163-196.

(Remarks on code availability)

Reviewer #1 (Remarks to the Author)

In this manuscript, Karjack et al. examine the relation between memory specificity and generalization in a group of school-aged children. Children watched videos of animals going home to different locations. Each location was nested within a place, which was nested within a town. Each animal was nested within a species, which was nested within a species class (mammal & bird). All places were home to one species, and all towns were home to one species class, in theory permitting participants to generalize a.) the town that would be home to new animals, b.) the type of place that would be home to a familiar animal, and c.) the place that would be home to a new exemplar from a specific species. The researchers tested children's generalization as well as their specific memory for the animal locations. They also examined and controlled for children's verbal intelligence and category knowledge. In line with their prior findings, the researchers found that generalization depended on specific memories, but the strength of the association between mnemonic specificity and generalization increased with age. The novel contribution of this particular study was examining generalization at two levels (e.g., to places and to towns). The higher level generalization was not dependent on mnemonic specificity, according to the specific analyses the researchers ran.

Comment 1: This manuscript has many strengths, but it is largely a conceptual replication of two prior studies by some of the same researchers (Ngo et al., 2021 & Buchberger et al., 2024).... However, the strengths of the task design replicate the strengths of the design of the task the authors used in prior studies. It was not clear to me that the addition of the higher level of generalization represents a significant conceptual advance above and beyond the authors' prior work, particularly because the hypotheses for high-level generalization were unclear. In particular, the authors ask: "First, can children generalize through extracting regularities across multiple instances and does this ability significantly improve with age? Second, is children' successful generalization dependent on memory specificity of the individual instances?" But both of these questions were already (beautifully!) addressed in Ngo et al. 2021.

We thank the reviewer for this comment. We agree that this work builds directly on our previous studies, and we appreciate the opportunity to clarify the important contributions of the current study. Our prior work focused on generalization at the level of individual object categories, where children and adults had to link a cartoon character (e.g., Moomin) to an object category (e.g., music instruments) based on four examples of Moomin collecting various music instruments. Here, the current study examines multi-level generalization and its dependency on memory specificity across different levels of abstraction. In fact, the ability to generalize across different levels of abstractions based on direct experiences is arguably the most powerful feature of generalization. Thus the current study extends our prior work in meaningful ways.

Moving beyond individual object categories, we considered two additional layers to the hierarchy of generalization. (1) intermediate-level generalization, requiring children to generalize to baby animals of the same species seen during encoding. This builds on prior findings (e.g., Hayes, 2007) suggesting that children can infer certain properties extending from parents to offspring, but our task required this inference in a more arbitrary spatial association context. And (2) higher-level generalization, requiring children to extend patterns from learned species to an unstudied species within the same class. This level of abstraction represents a more sophisticated form of generalization. We did not take for granted that 3-year-old children would succeed at this level.

In addition we extend the scope of generalization beyond the agent (e.g., animal or character) by showing that children also generalized to new places, demonstrating that knowledge extension is present for each constituent of the learned relationship. This represents another advance over prior designs, where generalization was only tested on one dimension.

We have now clarified these distinctions and added explicit predictions for each generalization level and research question on page 7 of the revised manuscript to better articulate the conceptual advances beyond our previous work.

Comment 2: Here, the authors additionally ask whether the relation between mnemonic specificity and generalization depends on the level of generalization. The authors note that the development of superordinate categorization emerges around 3 to 4 years. But the youngest participants in their samples were age 4. So it is not clear why the later emergence of superordinate vs. basic-level categorization should affect behavior in the age group they studied.

We have clarified in the Participants section of the revised manuscript that our sample includes 3-year-olds ($n=19$), alongside children aged age to 8 ($n_{4\text{-year-olds}}=23$; $n_{5\text{-year-olds}}=19$; $n_{6\text{-year-olds}}=19$; $n_{7\text{-year-olds}}=20$; $n_{8\text{-year-olds}}=21$). Thus our study does include children at the younger end of the age window during which superordinate-level categorization has been reported to be detected. As reported in the Supplemental Information section 3, even the youngest children in our sample performed above chance on the species-class category knowledge task, suggesting that they are generally aware of the broad taxonomic distinction between mammals and birds. However, the presence of superordinate categorization does not guarantee that children will make generalizations at that level. Indeed our findings suggest that generalization to novel species within a class, especially in an arbitrary spatial association context, improves significantly with age until middle childhood. More broadly, superordinate-level generalization can emerge at different points in development. Our use of the term was not intended to suggest a developmental onset, but to situate high-level within the broader literature on category abstraction.

Comment 3: In addition (and I describe this in more detail below), because of the structure of the task, higher-level generalization does not necessarily require the same number of specific memories or the same level of mnemonic precision that lower-level generalization requires. Thus, the lack of relation between memory specificity and high-level generalization seems like a potential feature of the design of this task, and not a general property of memory.

We agree that higher-level generalization in our task may not require the same number or precision of specific episodic memories as lower-level generalization. We view this as a meaningful and important feature of the task rather than a limitation. Our conceptualization of low- to high-level generalization reflects the hierarchical structure of category membership. By definition, broader (superordinate) categories encompass more members than its subcategories. As a result, generalization at the higher level is supported by a greater number of learning instances, which are distributed across a wider set of category members. This structure may lead to different demands on specific memories: higher-level generalization could rely on accumulated regularities, whereas lower-level generalization depends more directly on the retrieval of specific episodic details from a smaller set of experiences. From this perspective, the association between generalization and memory specificity at different levels is not a feature of our task design, but reflects a fundamental property of generalization as it operates over structured knowledge. This reflects how generalization functions in real-world learning contexts, where abstract inferences are often based on the integration of many experiences.

Comment 4: Related to this latter point, I also have some concerns and suggestions about aspects of the analytic approach and interpretation. I lay these out in more detail below. I believe that these concerns would be addressable in a revision; however, while I think addressing them would strengthen the manuscript, I still do not believe the conclusions the authors will be able to draw will be different from what they have already shown in two prior, published studies.

Throughout the manuscript, it would be helpful if the authors more explicitly stated what information / memories / knowledge would actually be required for each test of generalization, and analyze the data through this lens. As two examples: I was somewhat confused as to why the authors analyzed the relation between displacement accuracy and generalization in the New Animals task, rather than analyzing the relation between place memory accuracy and generalization. The authors do eventually run this analysis, and, unsurprisingly, find that displacement accuracy does not predict generalization above and beyond place memory accuracy. This is not surprising: To successfully generalize in this case, one only needs to know the place of the other animals, not their precise locations. Thus, it was unclear why displacement accuracy was the main focus.

We thank the reviewer for this helpful comment. We have now clarified the rationale of using each generalization task in the introduction and the methods (added texts are in blue font).

We placed an emphasis on displacement error because it reflects memory precision, as opposed to memory specificity categorically. Displacement error represents a more sensitive measure for memory precision, which is an important aspect of episodic memory characteristics (Harlow & Yonelinas, 2016; Murray et al., 2015; Richter et al., 2016). Further, using displacement error provides a higher-resolution measure that allows us to examine more graded relations between generalization and memory specificity. The parallel analyses using regional accuracy instead of displacement error showed the same results, which we now have now mentioned on page 28 of the revised manuscript. The full report on these analyses are in the IS section 6.

Comment 5: Similarly, I was confused by the dichotomization of memory specificity in the analyses examining why the relation between specificity and generalization increases with age. In particular, the authors report the finding that children are often able to successfully generalize despite having imperfect memories for the animal-place associations. This is potentially interesting, but again, given basic rules of logic, it doesn't seem like perfect memory is required for generalization. Instead, it seems like successful generalization would be possible in all these cases if participants remembered that two out of three of each animal lived in a specific place. In other words, the authors should clarify whether young children can generalize in the complete absence of relevant specific memories, or whether they are simply making inferences from 'noisier' and less accurate (but still accurate enough to be useful) specific memories. In fact, a very interesting follow-up experiment would be to 'train' adults on the basis of children's specific memories and then ask them to perform the generalization tasks. In other words, rather than providing adults with the standard learning phase, the researchers could yoke each adult participant to a child, and teach them their remembered animal locations. In this case, if adults' generalization appears comparable to children, then this would suggest that children are not necessarily relying on different types of memory representations or strategies to perform the generalization tasks, but rather that their less precise memories still permit generalization. The authors conclude that children are still able to generalize based on imprecise memories, but it seems like it's possible that anyone (regardless of age) could similarly generalize from these types of memories, and children just happen to have more imprecise memories. Indeed, from Figure 6, it seems like older children are actually better than younger children at generalizing from imprecise memories (e.g., for New Animals, older children successfully generalized in 59% of cases where they had imperfect memory compared to 38% for younger children). This is actually the comparison that seems most interesting, though it was not tested statistically. I'm not necessarily suggesting the authors run the yoked-adult study, but they should carefully consider their interpretation of the relation between specificity and generalization given that generalization still seems possible even on the basis of fewer accurate specific memories.

We thank the reviewer for the insightful comment. We agree that training adults or older children on the basis of younger children's specific memories would be a very interesting future follow-up. We conducted an additional set of analyses to better understand how different levels

of “imperfect” regional memory were associated with generalization accuracy. Here, we restricted our dataset to generalization trials where regional accuracy was imperfect (i.e., <9 across the 3 exemplars). We then calculated the number of exemplars (0, 1, or 2, out of 3) that had been correctly placed in their corresponding region and used this as a graded index of memory specificity (i.e., 0, 33%, or 66% levels of regional accuracy).

We conducted generalized linear mixed models for the New Animals, New Places, and Baby Animals tasks separately, specified as: `glmer(generalization accuracy ~ age * regional accuracy count + (1|participant))`

Figure S5. Distributions of the estimated probability of generalization accuracy on the trial-by-trial basis (*y*-axes) by an interaction between levels of imprecise memories (scaled, *x*-axes) and age in the New Animals (left), New Places (middle), Baby Animals (right) generalization tasks. On the top panel, every line is an individual participant. Color intensity represents age (measured in months). On the bottom panel, participants’ ages were grouped only for visualization purposes.

For New Animals generalization, we found that generalization accuracy was associated levels of regional accuracy, $\beta = 0.53$, $SE = 0.12$, $z = 4.29$, $p < .001$, and a trend towards significance with with age, $\beta = 0.29$, $SE = 0.17$, $z = 1.76$, $p = .08$, but no age * regional accuracy level interaction, $\beta = 0.12$, $SE = 0.12$, $z = 1.03$, $p = 0.30$.

For New Places generalization, the predicted probability of generalization accuracy increases with regional accuracy, $\beta = 0.69$, $SE = 0.12$, $z = 5.54$, $p < .001$, and interacted with age, $\beta = 0.32$, $SE = 0.12$, $z = 2.63$, $p = .008$. Age alone was not a significant predictor, $\beta = 0.04$, $SE = 0.16$, $z = 0.28$, $p = .77$.

For Baby Animals generalization, generalization accuracy also increased with levels of regional accuracy, $\beta= 0.41$, $SE= 0.12$, $z= 3.39$, $p< .001$, and also interacted with age, $\beta= 0.33$, $SE= 0.12$, $z= 2.68$, $p= .007$. The effect of age alone trended towards significance, $\beta= 0.26$, $SE= 0.15$, $z= 1.79$, $p= .07$.

These results, now included in Supplemental Information section 7, show that older children, compared to younger children, were better able to accurately generalize as the level of regional accuracy for a given animal species increased. These findings are indeed interesting and we would be happy to include them in our main text if the reviewer recommends it.

In addition, we explored the alternative operationalization of memory specificity by plotting the full range of regional accuracy score from 0 to 8 rather than percentages of correctly placed exemplars (see Figure below). However, we think that the percentage of correctly placed exemplars is a more relevant index of memory specificity in terms of its usage for successful generalization. For instance, a regional accuracy sum of 6 could result from a mixture of within-species regional accuracy patterns: 2, 2, 2, or 3, 2, 1. Therefore, it would not distinguish between consistent but incorrect group-level associations (all three horses placed at the wrong place in PearlCove) versus more fragmented but partially accurate memories (1 exemplar to the correct place and the other 2 exemplars elsewhere).

Figure caption: Distributions of the estimated probability of generalization accuracy on the trial-by-trial basis (y -axes) across the full range of regional scores for a given animal species (scaled, x -axes) in the New Animals (top), New Places (middle), Baby Animals (bottom) generalization tasks. On the left panel, every line is an individual participant. Color intensity represents age (measured in months). On the right panel, participants' ages were grouped only for visualization purposes.

However, as we stated in the originally submitted manuscript, our goal in examining mismatches between generalization and memory specificity was to understand *why* the contingency between generalization and memory specificity was weaker in younger children compared to older children. Specifically, we asked whether this weaker association was driven by (1) a failure to generalize even when specific memories were accurate (i.e., a knowledge-to-behavior gap, or (2) successful generalization despite unreliable specific memories. It is very likely that both factors are at play, but their relative contributions to lower coupling between generalization and memory specificity at the younger age was unclear. Importantly, this question was not addressed in prior studies. Our original analyses and the revised analyses clearly showed that the weaker association between generalization and memory specificity in younger children was primarily driven by accurate generalization coupled with unreliable memory specificity of the individual episodes.

The goal was *not* to compare whether children across the age range could equally generalize with the same level of memory specificity. In fact, the analyses that we performed above to respond to the reviewer's comment clearly showed that with age, greater memory specificity (albeit imprecise) led to better generalization.

Comment 6: Relatedly, in addition to avoiding the binary division of perfect vs. imperfect regional memory, the authors should also avoid the arbitrary binary division of age group.

We thank the reviewer for this helpful comment. In response, we have revised our contingency analyses and updated our figures to characterize the effects of age (measured in months) as a continuous variable. However, for the purpose of comparing the frequencies between two generalization-specificity mismatching cases (under **Unpacking generalization-specificity contingencies**), we needed to treat regional accuracy as either reliable/perfect or not. In our view this was the most appropriate approach to characterize why generalization and specificity coupling is lower for younger children. We therefore compared the frequencies with which children accurately generalized without imperfect regional accuracy versus inaccurate generalization with perfect regional accuracy. This analysis clearly showed that when there is a mismatch, younger children, but not older children, showed greater frequency of accurate generalization when their specific memories are not unreliable. As stated in our response to your comment 5, we have conducted complementary analyses that examined generalization accuracy across a continuum of regional accuracy levels in Supplemental Information (Figure S6), providing a more fine-grained analysis of how generalization scales with different degrees of memory specificity across ages.

Comment 7: Similar to my first comment, it was not clear to me why the authors used displacement error in their analysis of generalization in the New Species task. It is not surprising here that displacement accuracy does not predict generalization, because remembering the specific locations of animals is unnecessary to perform this task successfully. It seems as though to successfully generalize in this task, participants need to

simply remember that the majority of mammals came from Town A and the majority of birds came from Town B. Presumably the researchers can compute whether participants remembered this information based on participants' placement of the animals in the location memory test. Again, as stated in my initial evaluation of the manuscript, it is unclear what the lack of relation between specificity and generalization here really tells us about memory development, because highly specific memories do not seem required to perform this generalization task. It seems interesting to determine whether children are performing this generalization task by drawing on imprecise specific memories OR by drawing on some type of aggregate / generalized / incrementally updated representation that is not dependent on individual episodes at the time of retrieval, but it does not seem like the authors have the data to disambiguate these two possibilities here.

Please also see our response to the reviewer's comment 4. The findings on the contingency between generalization and memory specificity are the same when we used regional accuracy as a predictor instead of displacement error, which we have now noted on page 28 of the revised manuscript. These parallel results with regional accuracy are now reported in the SI section 6. Displacement error represents a more sensitive measure for memory precision, which is an important aspect of episodic memory characteristics (Harlow & Yonelinas, 2016; Murray et al., 2015; Richter et al., 2016). Therefore the examination of (i) the development of memory precision, and (ii) its relationship to generalization, benefit from our ability to quantify children's memories' absolute deviation from the learned events.

Comment 8: I am concerned that the authors did not include random participant slopes in any of their models. The inclusion of intercepts only may be inflating the Type I error rate.

We appreciate the reviewer's suggestion regarding the inclusion of random participant slopes. We re-ran all models in the contingency analyses with random slopes for displacement error, in addition to the existing random intercepts in the original models. The inclusion of random slopes yielded the same results. We have updated our reported statistics from the revised model specification.

Minor:

Comment 9: Why was the regional accuracy score manually coded as opposed to derived from the location coordinates of the animal placement?

The regional accuracy scores could be derived from the location coordinates, but we scored them live as it was significantly less logistically and technically complex.

Comment 10: While I eventually understood the authors' intended meaning, it would be helpful to clarify what "accurate generalization in the face of unreliable animal-place associations" means upon first mention. Initially, I was confused about whether 'accurate generalization' here means accurate generalization given incorrect beliefs. (i.e., the participant selected the wrong answer but 'generalized' correctly — they incorrectly remember that most horses live in the house and then correctly extend that incorrect belief to novel horses) or accurate generalization given the true structure of the task.

We thank the reviewer for pointing out the potential mis-interpretation. We have now added more details about the generalization and memory specificity mismatch cases on page 31.

There were two sources of mismatch between generalization and memory specificity: (1) successful generalization despite unreliable memory specificity, and (2) failed generalization despite reliable memory specificity. In the first case, children accurately generalized when their animal-place associative memories were not entirely reliable. For example, a child may correctly infer that a new horse would go to the castle, even when they had not remembered that the three learned horses were originally placed there. In the second case, children failed to generalize, even though their memory for the original animal-place associations were accurate. That is, a child may have remembered that all three learned horses had gone to the castle, but failed to apply that pattern to a new horse, a baby horse, or a new castle.

Comment 11: Although it didn't detract from the manuscript, the purpose of the baby animals generalization task was not clear to me.

We have revised the Introduction to better explain how each task corresponds to a different level of generalization. The Baby Animals task is conceptualized as an intermediate level of generalization. It requires children to extend learned regularities to new exemplars from the same basic-level category (e.g., horses), but where the test stimuli are less perceptually similar to the adult exemplars encountered during encoding. This knowledge extension is, closer to the directly learned experiences than the New Species class, but more abstract than applying patterns to similar new adult exemplars, as in the New Animals task. This task was also inspired by previous research on inductive reasoning that tested whether participants would infer that a baby offspring of a mythical animal would also contain the same unobservable biological property (e.g., containing sarca cells) as the probed animal (e.g., Hayes, 2003).

(Remarks on code availability)

The code is available and clearly commented, but I have not looked at it closely or tried to run it.

Reviewer #2 (Remarks to the Author)

The manuscript by Sabrina Karjack, Nora S. Newcombe, and Chi T. Ngo addresses important questions, namely how children manage to generalize, and how the role of episodic memory in this process changes with age. This topic is both timely and not well understood, making the study a valuable contribution. The study design is rigorous and effectively investigates various levels of abstraction, while also capturing episodic memory performance in children aged 3 to 8 years.

The findings suggest that higher-level generalization in children is not reliant on detailed, specific memories. On the other hand, lower-level generalization shows a dependence on specific, detailed memories, and that this reliance increases with age.

While the manuscript is well written and insightful, there are some areas that would benefit from further clarification.

Comment 1: In the abstract, the phrase "...characteristics of individuals..." seems to imply individuals as in persons/beings. The authors might have meant individual experiences or episodes. Can the authors please clarify?

We meant persons/beings and have now changed this phrase to "characteristics of individual people" for clarity.

Comment 2: There are several jargons in the abstract that are used interchangeably without definitions. These are memory specificity/memory precision, and abstraction/generalization/regularities. Revising the use of these terms will increase readability.

We thank the reviewer for this comment. We have now revised the abstract such that key terms are now better introduced and defined in the introduction.

Comment 3: It would be helpful if the authors could provide the study hypotheses alongside the research questions. Was the study preregistered?

This study was not pre-registered. We have stated this at the end of our introduction. We have also revised the current study section of the introduction to include clear hypotheses corresponding to each research question.

Comment 4: Could the authors provide a more detailed breakdown of the ages of the excluded participants? Providing descriptive statistics of the PPVT task (with age) will also be helpful.

We have added a plot in the Supplemental Information (Figure S1) highlighting the excluded participants. We also included the descriptive statistics of the PPVT task in the figure caption.

Comment 5: . In the description of the typicality of the task stimuli, it is stated that "Within each animal species, half of the animals were more typical, whereas the other half were less typical." Could the authors clarify what the typicality was based on? Was it determined using previous typicality norms with children (e.g., Posnansky, 1978)?

We have revised the information in our materials section to clarify that all of the animals used as stimuli in this study were selected from two normed categories of birds and mammals from a previous study (Posnansky, 1978). From these lists, we selected the typical versus atypical animals from the higher and lower ends of typicality, respectively. The lists of animals separated by typicality are now reported in Table 1 in the Supplemental Information.

Comment 6: It is unclear from the manuscript whether any correction for multiple comparisons was applied. Given the repeated testing of similar models on related measures (of generalization), should this be taken into consideration?

We did not correct for multiple comparisons. We conducted hypothesis-driven analyses and therefore avoided correcting for multiple comparisons which may lead to Type II error.

Comment 7: Can the authors clarify how displacement errors were included in the various models in Section 2.1? For the new animals/new places/baby animals models, were displacement errors aggregated within each animal species? For the model of New Species (p. 22), did the authors examine whether displacement errors of same-class species are actually correlated? The reasoning here is that displacement error in this model was treated differently from the other models. Given the central role of the findings in this section to the study, it would be important if the authors could verify to what extent displacement error as a predictor is comparable in the measurement properties across the lower- vs. higher-level generalization tasks.

We thank the reviewer for this helpful comment. For the New Animals, New Places, and Baby Animals models, the displacement error was computed by averaging the errors across the three exemplars of a given animal species (e.g., horse 1, horse 2, horse 3) for each generalization trial. For the New Species model, displacement error was computed as the average error across all animals of species that belonged to the same species class (e.g., mammals) that were assigned to a given town during encoding. This included 12 trials: 3 exemplars x 4 species (e.g., horses, cats, dogs, and lions for mammals). By design, there are more trials that belong to the same species class compared to those that belong to the same species - as this difference is inherent to any hierarchical structured knowledge. We recognize the importance of ensuring that

displacement error as a predictor has comparable measurement properties across tasks. To this end, we conducted a Levene's test for homogeneity of variance across the four generalization tasks, which showed no significant differences in variance, $F(3, 6461) = .57, p = .64$, indicating that displacement error was equally variable across tasks. We would be happy to report these statistics and/or the figure below in the Supplemental Information if the reviewer thinks that this would strengthen the manuscript.

Regarding the reviewer's question about the New Species model and "whether displacement errors of the same-class species actually correlated for the model of New Species (p. 22)", can the reviewer clarify what they meant by this question? We did not understand what type of correlation the reviewer had in mind.

Comment 8: Related to the question above, in Figure 5, it looks like the range of New Species generalization success (the y-axis) is more restricted compared to the other generalization tasks. Could this have contributed to the lack of relationship with displacement error in this task?

While the New Species task does have a different distribution of scores compared to the other tasks, the range of generalization accuracy does, in fact, span the full 0 - 1 scale, as in the other generalization tasks. We thank the reviewer for flagging the discrepancies in axes labels, and have revised Figure 5 so that the y-axes are consistent across the panels for clearer visual comparison.

To test whether the lack of significant association between displacement error and generalization accuracy in the New Species task may be attributable to the restricted variance, we examined the distribution of generalization accuracy across all four tasks (see Figure below). A Levene's test showed no significant difference in variance, $F(3, 480) = 1.05, p = .37$, indicating that generalization accuracy was equal across tasks. Combined with our response about the

displacement error variance above, it is unlikely that the null finding in the New Species task is due to the restricted ranges in variances.

Comment 9: Did the new species stimuli also differ by typicality? Did this affect performance on New Species generalization?

We thank the reviewer for bringing up this question. The New Species task stimuli in blocks A and B differed in typicality. Typicality significantly impacted the New Species generalization task, and it interacted with age. The age-related increase in generalization accuracy was greater for typical species compared to atypical ones. The findings suggest that conceptual familiarity may facilitate higher-level generalization in younger children. We have now included typicality as a fixed effect in the models predicting generalization accuracy, alongside age and verbal intelligence (see page 21).

Comment 10: While I appreciate the rationale and potential insights offered by Section 2.2, it is difficult to follow what kind of analysis was done to compare the red and blue circled cells of Figure 6. Specifically, as the denominators of these cells are different, is it meaningful to directly compare them to each other statistically (if this was done)? Some elaboration on the statistical comparison would be helpful.

We thank the reviewer for this comment. In our initial submission, the conditional probability analyses had limitations and we have now addressed the concern with more straightforward and interpretable analyses to better investigate the age-related differences in the generalization - specificity mismatch. Previously, our analyses binned the frequencies of responses by age groups, which did not account for participant-specific frequency patterns or the examination of age as a continuous. In our revised analyses, we approached this question by calculating the

within-participant proportion of trials that fell into two generalization-specificity mismatch types: (1) inaccurate generalization despite perfect regional accuracy, and (2) accurate generalization despite imperfect regional accuracy. We then conducted linear mixed models predicting the proportion of trials from mismatching types and age (measured in month as a continuous variable), and their interaction. The model was specified as:

$$\text{lmer}(\text{proportion of trials} \sim \text{mismatch type} * \text{age} + (1 | \text{participant}))$$

These results are now reported under the Unpacking the age-related increase in the generalization-specificity contingencies section of the Results. These analyses showed a significant mismatch type x age interactions for three tasks. Younger children showed a significantly higher proportion of trials where they generalized accurately despite imperfect memory, compared to the reverse mismatch type. However, this asymmetry diminished with age.

Comment 11: In the discussion, the authors stated, “This set of results suggests that from early to middle childhood, children were able to recognize and apply higher-level patterns between the species classes and towns, even when they failed at the lower-level associations between species and specific places.” However, the manuscript does not explain why this might be the case. Could the authors elaborate on what aspects of cognitive or brain development may contribute to the early onset of this ability or the formation of “summary” representation?

Given the limited existing research on multi-level generalization in both children and adults, we can only speculate on why high-level generalization appears to be less reliant on memory for individual episodes. One possibility is that high-level generalization reflects a form of implicit learning whereby children aggregate broad patterns across learning episodes without necessarily being aware of or able to recall individual instances. aggregated across the learning phase. Prior work has shown that complex relational structures are often acquired implicitly, that is, unintentionally and without awareness (Cleeremans et al., 1998; Garvert, Dolan, & Tim Behrens, 2017; Reber, 1989, Seger & Augart, 1994). Such learning mechanisms may support the formation of summary representations over time. At the neural level, it is possible that lower-level generalizations directly call upon the retrieval of specific episodic memories via hippocampal-dependent mechanisms, either via the trisynaptic pathway (Shapiro et al., 2017) in the hippocampus, or through the hippocampal big loop (Koster et al., 2019). More abstracted representations of the structured knowledge may be built up in the neocortex or via the mono-synaptic pathway within the hippocampus, throughout learning and are less reliant on the retrieval of specific instances.

Cleeremans, A. Destrebecqz, & Boyer, M. (1998). Implicit learning: news from the front. *Trends in Cognitive Sciences*, 2, 406-416.

Garvert, M. M., Dolan, R. J., & Behrens, T. E. J. (2017). A map of abstract relational knowledge in the human hippocampal-entorhinal cortex. *Elife*, 6, e17086.

Reber A. S. (1989). Implicit learning and tacit knowledge. *Journal of Experimental Psychology: General*, 118, 219-235.

Seger, C. A., Augart, C. (1994). Implicit learning. *Psychological Bulletin*, 115, 163-196.

Reviewer #3 (Remarks to the Author)

Reviewer #3 (Remarks to the Author):

The present study assessed generalization across different levels of abstraction and relations with memory specificity in 128 children ages 3-8 years old. Generalization across all levels improved with age. While higher-level generalization was not related to memory specificity, lower-level generalization was such that the relationship between generalization and memory specificity strengthened with age. Younger children seem to be able to generalize despite imprecise episodic memories. This manuscript is clearly written and addresses several gaps in knowledge regarding generalization in children. Below are some comments and questions that seek to clarify the motivations for aspects of the study design, details relating to the results, and interpretations of the findings.

Comment 1: -The hypotheses were not clear - there was not a clear motivation for including each level of abstraction that was tested in this study. Were differences in the relation between memory specificity and generalization at different levels of abstraction expected? What was the rationale for including each of the 4 generalization tasks? These points could have been addressed more clearly in the introduction and methods.

We thank the reviewer for this comment. We have revised the Introduction to include specific hypotheses and rationale for designing the different generalization tasks in our study. In addition, we have also added a conceptual figure (Figure 1) to illustrate the hierarchical levels of generalization and how they map onto the four generalization tasks (see also our response to Reviewer 1's comment 1). And finally, we have also added a description on what each generalization task should reflect in the methods section from pages 14 to 17.

Comment 2: Related to the previous point, the animal-location recall task also seems to not only be a more difficult task (not forced choice) but also a more difficult motor movement. Are there any data that speak to the youngest children's ability to accurately move the animal to their desired position? This seems important for interpretations regarding memory specificity.

We agree with the reviewer that forced-choice formats are generally easier than recall tasks. However, we were particularly interested in measuring children's *memory precision with an absolute distance between the child's recalled position and the true location, as an index of episodic memory capacity*. This continuous measure offers a sensitive method to quantifying memory error magnitude that is not possible with categorical or forced-choice formats. In our view, this aspect of our work advances the understanding of episodic memory capacity in children, as the most prior research has focused on the types of memory errors rather than the degree of memory error. Figure 6B (left) illustrates the range in recall precision on a trial-by-trial level when we used displacement error to index memory specificity. We have explicitly stated our motivation for measuring memory precision in the introduction of the revised manuscript (page 4).

Both aspects of episodic memory – remembering what happened and where it occurred, and the precisions of such memories (Harlow & Yonelinas, 2016) – improve from early to middle childhood (reviewed in Newcombe et al., 2023). Many studies have shown that children's abilities to retrieve associative memories between places and objects or actions improved from ages 4 to 8 or 9 (Bauer et al., 2012, Picard et al., 2012). However, we know far less about memory precision in early development, as only a handful of studies have quantified memory precision as the deviation between true and retrieved memories in young children. These studies found that memory precision for the specific objects' positions within a spatial array improves from ages 3 to 7 (Lambert et al., 2017) or even into adulthood (Peng et al., 2023; Schommartz et al., 2024). In sum, younger children are less able to bind items to locations, and reconstruct the item-position associations with less precision.

We acknowledge that inter-individual or age-related differences in fine motor control could influence performance on this task. To minimize this confounded factor, the animal-location recall task was done on a laptop with an external mouse which allowed most children to drag and drop the animals precisely where they wanted. If a child struggled with using the mouse, the experimenter instructed them to touch the screen at the desired location and hold their finger there until the animal was placed by the experimenter. The experimenter then confirmed the dropped locations with the child multiple times to ensure that they accurately reflected the child's intended recall before proceeding to the next trial. We have now added this description to the methods section on page 17 of the revised manuscript.

Comment 3: Several aspects of the results section warrant further clarification. Correlation values for stats in lines 361-364 should be stated to get a sense of the effect sizes. Result statistics assessing the effect of age on new species generalization task (lines 382-384) are missing along with model set up details (was species class category knowledge an additional predictor to PPVT?). What statistical tests are used in assessing generalization-specificity mismatch (Figure 6)?

We thank the reviewer for this suggestion. We have now improved the reporting of the results in two ways. First, we have now added the Cohen's f effect sizes for all correlations and the standardized coefficient (and standard error) for each predictor in our mixed models, reported in the Results section. Second, we have added all model specifications in each set of analysis.

With regards to the conditional probability analyses reported in our initial submission, please see our response to Reviewer 2's Comment 10.

Comment 4: Was the effect of age on generalization accuracy tested while considering both species class category knowledge and PPVT for all generalization tasks? Better knowledge of species class category could help participants at the least in the new and baby animals task.

Verbal ability, as measured by PPVT, was included in *all* memory performance analyses, including all generalization and memory specificity models, and we report these effects throughout the Results. The species class category knowledge, which estimates children's abilities to group mammals with other mammals, and birds with other birds, at least with our stimulus set, was included in the New Species model. In contrast, the New Animals, New Places, and Baby Animals tasks required generalization within a species animal species rather than between different species across classes. Therefore we did not include the species class knowledge as a predictor in those models.

Comment 5: Based on the intercepts of the lines of fit in Figure 2, the youngest participants appeared to be, on average, above chance on the new animals, new places, and baby animals tasks (where chance is 25%) and below chance on the new species task (where chance is 50%). Could this (or the compressed range/decreased variation in performance for the new species task) have contributed to the lack of age by displacement error effect on this task versus the others, especially given that effects seem to trend in the same direction as the other generalization tests? Some caution in interpretation seems warranted.

We thank the reviewer for this comment. Please see our responses to Reviewer 2's comments 7 and 8. In short, we tested for equal variance in generalization accuracy and in the displacement error included in the models across the four generalization tasks and indeed showed that there is equal variance among them. Therefore it is unlikely that the non-significant relationship between generalization accuracy in the New Species task and displacement error is due to restricted variances.

Comment 6: Regarding the generalization-specificity mismatch, could the lack of effect for older children be due to ceiling performance? It looks like for most trials, older children have good specificity, so there are very few trials available to assess potential differences

between the two generalization-specificity mismatches. Additionally, conditioning by specificity, it appears that for both younger and older children accurate specificity implies better generalization. Conversely, inaccurate specificity seems to imply worse generalization for younger children while being slightly better for older children (bottom rows in Figure 6). Does this not provide evidence that younger children do indeed rely on memory specificity for generalization contradicting the third conclusion as stated in the discussion section? That is, when memory specificity is worse, younger children generalize worse whereas older children do not. If younger children indeed did not rely on memory specificity relative to older children, their generalization when memory specificity was low should be relatively higher compared to older children which doesn't appear to be the case.

We agree with the reviewer that the ceiling effects among older children, particularly those aged 7 and 8 may limit our ability to detect differences between two generalization-specificity mismatch types. Nonetheless, the pattern with the younger ages, 3, 4, 5, and 6 clearly shows an increasingly greater prevalence of accurate generalization paired with unreliable memory specificity with decreasing age. We have added a cautionary note regarding the ceiling effect in both the Results section (pages 33 and 34) and the Discussion (page 42) of the revised manuscript.

With respect to the reviewer's second point, please see reviewer 1's comment 5 for our revised analyses on the relationship between generalization and levels of memory specificity as a function of age. Children, regardless of their ages, rely on memory specificity to generalize. The key finding is that the coupling of generalization and memory specificity is weaker among the younger ages. We never stated that younger children did not rely on memory specificity, only that they did to a lower degree. Our third conclusion in the discussion focused on the different sources that contribute to the mismatch between generalization and memory specificity. There is an asymmetry in younger children indicating that they are more likely to generalize accurately despite unreliable memory specificity rather than the reverse case. This asymmetry diminishes with age. On page 42, we stated that "However, it was *not* the case that preschoolers had memories for individual episodes but failed to apply them to generate novel inferences. Our findings suggest that the opposite is true: younger children could still make appropriate generalizations based on imprecise episodic memories." This conclusion does not imply whether younger children's generalization would be higher or lower compared to older children when memory specificity is low.

Minor:

Comment 7: Given the possible reliance on audio cues, it would be useful for a reader to know whether all participants were native English speakers.

English proficiency was one of the inclusion criteria for this study. All recruited participants met this criterion. We have now included the full list of inclusion criteria in the Participant section of the Methods.

Comment 8: Figure 1B shows an overview of the world state after a block, whereas the text implies that participants were shown this world state ‘sequentially’ where each animal disappeared after moving to its respective home location. Were the participants ever shown the overview in Figure 1B? I think not, but it would be useful to explicitly state so. If they were shown, potential influence of the displayed complete world state on subsequent performance should be discussed.

Participants were never shown the overview seen in Figure 1B. This figure was only used for illustrative purposes for the reader. We thank the reviewer for bringing up this potential point of confusion and we have now clarified this in our figure caption.

Comment 9: Participants are presented with animals with low and high typicality however data appears to be analyzed after combining across both. Are there any meaningful differences in generalization performance between high vs low typicality?

We thank the reviewer for asking this question. We have revised our analyses in the results (section 1. Age-related improvements in generalization) to examine the influence of typicality on generalization accuracy, in addition those by age and verbal intelligence. Instead of simple regressions reported in the first version of the manuscript, we conducted a linear mixed model for each generalization task with age*typicality interaction and verbal intelligence as fixed effects, and participants as a random intercept.

$$\text{lmer}(\text{accuracy} \sim \text{age} * \text{typicality} + \text{verbal intelligence} + (1 | \text{participant}))$$

Interestingly, typicality significantly impacts generalization performance on the New Species task, and it interacts with age. The age-related increase in generalization accuracy is greater for typical, compared to atypical, species. Typicality did not impact the other generalization tasks.

Reviewer #3 (Remarks on code availability):

An access code was required to view the code

We apologize for this oversight. We have now removed permission requests to access task and analysis scripts.

Reviewer #4 (Remarks to the Author):

Reviewer #4 (Remarks on code availability):

Code link lead to a request for access.

We have now removed permission requests to access task and analysis scripts.

Response to Reviewer's Comments

Reviewer #1 (Remarks to the Author):

My previous assessment of this manuscript was that it was largely methodologically sound, but potentially not that novel given how closely related it is to the authors' previous work. I believe the authors have done a nice job of responding to my methods concerns, and I think the new analyses and figures are helpful in understanding the complex relations between memory specificity and generality. I believe this work has been done very well, even more so after the revision. That said, given that generalizing more broadly logically does not require detailed memories, I still do not believe that the additional level of hierarchy necessarily extends our understanding of memory development that much above and beyond existing work.

Response: We thank the reviewer for their thoughtful feedback, which has significantly improved our manuscript. We agree that broad generalization, as demonstrated empirically in this work, does not necessarily rely on specific memories. Existing theories of memory generalization often distinguish between different types—such as similarity-based (perceptual) and relational (associative)—but typically do not address the possibility of **multiple levels of abstraction** within each type, nor how these levels relate to memory specificity (Taylor et al., 2021). Our findings show that, from early to middle childhood, high-level generalization can occur independently of specific memory retrieval. These results highlight the need for models of memory generalization to account for varying degrees of abstraction and their developmental relationship to memory specificity.

Taylor, J. E., Cortese, A., Barron, H. C., Pan, X., Sakagami, M., & Zeithamova, D. (2021). How do we generalize? *Neurons, Behavior, Data Analysis, and Theory, 1*.
<https://doi.org/10.51628/001c.27687>

Reviewer #3 (Remarks to the Author):

The authors have largely addressed our comments and questions.

We appreciate the updated analyses conducted to assess the age-related differences in the coupling of memory specificity and generalization performance. In Figure S5, it would be useful to show where the curve and model statistics extend for the cases with 100% memory specificity as well. For example, if older kids' generalization stays as it is at 100% specificity or if younger kids' generalization gets a non-linear boost at 100% specificity, the coupling strength by age effect could go away. Generally the authors should also consider

the lack of age by regional accuracy interaction in the new animals task in their conclusion. Could the authors please clarify?

Response: Figure S5 indeed shows the cases with 100% memory specificity. Could the reviewer please rephrase to help us better understand the suggestion?

With regards to the comment about “the authors should also consider the lack of age by regional accuracy interaction in the new animals task in their conclusion”, we have previously reported a significant age x regional accuracy interaction in the New Animals Task (See SI. 6.

Contingencies of generalization on regional accuracy (instead of displacement error):

For the New Animals generalization task, generalization accuracy was associated with verbal intelligence ($\beta= 0.37, p= .02$), age ($\beta= 1.09, p<.001$), regional accuracy ($\beta= 1.11, p<.001$), and an age*displacement error interaction ($\beta= 0.42, p= .004$). The likelihood of correctly generalizing that a new animal exemplar would go to the place as its same-species members depended on children’s memory accuracy of animal-location associations for all children. However, the degree of this contingency varied with age: older children showed a stronger tie between generalization and memory accuracy compared to younger children. Nonetheless, younger children’s (ages 3-5) generalization accuracy was significantly associated with regional accuracy ($\beta= 1.26, p<.001$).

Reviewer #3 (Remarks on code availability):

The code is clear and annotated.

Response: Thank you for reviewing our code.

Reviewer #4 (Remarks to the Author):

Reviewer #5 (Remarks to the Author):

The authors have successfully addressed most of my comments. However, I believe the Discussion section would benefit from including the information raised in Comment 11, as it touches on potential mechanisms underlying the key findings. Although this aspect is not central to the manuscript’s primary goals, its inclusion would provide a more comprehensive interpretation of the results within the context of the broader neuroscientific literature.

Comment 11: In the discussion, the authors stated, “This set of results suggests that from early to middle childhood, children were able to recognize and apply higher-level patterns between the species classes and towns, even when they failed at the lower-level associations between species and specific places.” However, the manuscript does not explain why this might be the case. Could the authors elaborate on what aspects of cognitive or brain development may contribute to the early onset of this ability or the formation of “summary” representation?

Response: We thank the reviewer for this suggestion and have now added the text to our discussion section. “Given the limited existing research on multi-level generalization in both children and adults, we can only speculate on why high-level generalization appears to be less reliant on memory for individual episodes. One possibility is that high-level generalization reflects a form of implicit learning whereby children aggregate broad patterns across learning episodes without necessarily being aware of or able to recall individual instances. aggregated across the learning phase. Prior work has shown that complex relational structures are often acquired implicitly, that is, unintentionally and without awareness (Cleeremans et al., 1998; Garvert, Dolan, & Tim Behrens, 2017; Reber, 1989, Seger & Augart, 1994). Such learning mechanisms may support the formation of summary representations over time. At the neural level, it is possible that lower-level generalizations directly call upon the retrieval of specific episodic memories via hippocampal-dependent mechanisms, either via the trisynaptic pathway (Shapiro et al., 2017) in the hippocampus, or through the hippocampal big loop (Koster et al., 2019). More abstracted representations of the structured knowledge may be built up in the neocortex or via the mono-synaptic pathway within the hippocampus, throughout learning and are less reliant on the retrieval of specific instances.”